# Identification of an H-Ras nanocluster disrupting peptide
Candy Laura Steffen [1,8], Ganesh babu Manoharan [1,8], Karolina Pavic[1], Alejandro Yeste-Vázquez[2,3], Matias Knuuttila [4], Neha Arora[5], Yong Zhou [5], Harri Härmä[6], Anthoula Gaigneaux [7], Tom N. Grossmann [2,3] & Daniel Kwaku Abankwa [1,4 ✉]

Hyperactive Ras signalling is found in most cancers. Ras proteins are only active in membrane nanoclusters, which are therefore potential drug targets. We previously showed that the nanocluster scaffold galectin-1 (Gal1) enhances H-Ras nanoclustering via direct interaction with the Ras binding domain (RBD) of Raf. Here, we establish that the B-Raf preference of Gal1 emerges from the divergence of the Raf RBDs at their proposed Gal1-binding interface. We then identify the L5UR peptide, which disrupts this interaction by binding with low micromolar affinity to the B- and C-Raf-RBDs. Its 23-mer core fragment is sufficient to interfere with H-Ras nanoclustering, modulate Ras-signalling and moderately reduce cell viability. These latter two phenotypic effects may also emerge from the ability of L5UR to broadly engage with several RBD- and RA-domain containing Ras interactors. The L5UR-peptide core fragment is a starting point for the development of more specific reagents against Ras-nanoclustering and -interactors.

Ras is a major oncogene and recent advances in its direct targeting have validated its high therapeutic significance[1,2]. The three cancer-associated Ras genes encode four different protein isoforms: K-Ras4A, K-Ras4B (hereafter K-Ras), N-Ras, and H-Ras. These membrane-bound small GTPases operate as switchable membrane recruitment sites for downstream interaction partners, called effectors. Downstream of mitogen and growth factor sensing receptors, inactive GDP-bound Ras is activated by guanine nucleotide exchange factors (GEFs), which facilitate GDP/ GTP-exchange[3,4]. The two switch regions of GTP-Ras undergo significant conformational changes upon activation, thus enabling binding to the Ras binding domain (RBD) or Ras association (RA) domain of effectors, such as Raf, PI3Kα, and RASSF proteins. These effectors are implicated in cell proliferation, growth, and apoptosis, respectively[5,6].

Current evidence suggests that Ras proteins promiscuously interact with any of the three Raf paralogs, A-, B- and C-Raf. Raf proteins reside as autoinhibited complexes with 14-3-3 proteins in the cytosol and are activated by a series of structural rearrangements that are still not understood in full detail[7,8]. The first crucial step is the displacement of the RBD from the cradle formed by the 14-3-3 dimer[7]. Simultaneous binding of Ras and 14-3-3

to the N-terminal region of Raf is incompatible due to steric clashes and electrostatic repulsion, which is only relieved if the RBD and adjacent cysteine-rich domain of Raf are released from 14-3-3 for binding to membrane-anchored Ras. Allosteric coupling between the N-terminus of Raf and its C-terminus then causes dimerization of the C-terminal kinase domains, which is necessary for their catalytic activity[8–10].

The Ras-induced dimerization of the Raf proteins requires di-/oligomeric assemblies of Ras, called nanoclusters[11]. Initially it was estimated that 5–20 nm sized nanoclusters contain 6–8 Ras proteins and that nanoclustering was necessary for MAPK-signal transmission[12–14]. More recent data revealed that nanoclusters are dominated by Ras dimers[11,15]. Intriguingly, Ras nanoclustering can be increased by Raf-ON-state inhibitors that induce Raf dimerization and increase Ras–Raf interaction, suggesting that Raf dimers are integral components of nanocluster[16,17]. The reinforced nanoclustering may thus contribute to the paradoxical MAPK-activation that is observed with these inhibitors[18].

Currently, less than a dozen proteins are known that can modulate Ras nanoclustering[19]. These proteins do not share any structural or functional similarities, suggesting that their mechanisms of nanocluster modulation

[1]Cancer Cell Biology and Drug Discovery group, Department of Life Sciences and Medicine, University of Luxembourg, 4362 Esch-sur-Alzette, Luxembourg. [2]Department of Chemistry and Pharmaceutical Sciences, VU University Amsterdam, Amsterdam, The Netherlands. [3]Amsterdam Institute of Molecular and Life Sciences (AIMMS), VU University Amsterdam, Amsterdam, The Netherlands. [4]Turku Bioscience Centre, University of Turku and Åbo Akademi University, 20520 Turku, Finland. [5]Department of Integrative Biology and Pharmacology, McGovern Medical School, UT Health, Houston, TX 77030, USA. [6]Chemistry of Drug Development, Department of Chemistry, University of Turku, 20500 Turku, Finland. [7]Bioinformatics Core, Department of Life Sciences and Medicine, University of Luxembourg, 4367 Esch-sur-Alzette, Luxembourg. [8]These authors contributed equally: Candy Laura Steffen, Ganesh babu Manoharan. ✉e-mail: daniel.abankwa@uni.lu

are diverse. The best understood nanocluster scaffold is the small lectin galectin-1 (Gal1), which specifically increases nanoclustering and MAPK-output of active or oncogenic H-Ras[20–22]. Consistently, upregulation of galectins has been linked to more severe cancer progression[23]. For many years, it was mechanistically unclear, how this protein that is best known for binding β-galactoside sugars in the extracellular space affects Ras membrane organization on the inner leaflet of the plasma membrane[24,25]. While it was first suggested that the farnesyl tail of Ras is engaged by Gal1[26], it was later on shown that neither Gal1 nor related galectin-3, which is a nanocluster scaffold of K-Ras, bind farnesylated Ras-derived peptides[27,28].

We previously proposed a model of stacked dimers of GTP-H-Ras and Raf as the minimal unit of active nanocluster that can be further enhanced by Gal1[29]. We confirmed that Gal1 does not directly interact with the farnesyl tail of Ras proteins, but instead engages indirectly with Ras via direct binding to the RBD of Raf proteins ($K_D = 106 \pm 40$ nM)[29]. Given that Gal1 is a dimer at low micromolar concentrations in cells ($K_D = 7$ μM)[30,31], we hypothesized that dimeric Gal1 stabilizes Raf-dimers on active H-Ras nanocluster. In line with this, in particular B-Raf-dependent membrane translocation of the tumor suppressor SPRED1 by dimer inducing Raf-inhibitors was emulated by expression of Gal1[32]. Our mechanistic model suggests that dimeric Gal1 stabilizes the dimeric form of Raf-effectors downstream of H-Ras. This enhances H-Ras/ Raf signaling output, not only

by facilitation of Raf-dimerization, but also by an allosteric feedback mechanism that enhances the nanoclustering of H-Ras. Altogether, a transient stacked dimer complex of H-Ras, Raf and Gal1 is formed, which also shifts the H-Ras activity from the PI3K to the MAPK pathway[29]. Current galectin inhibitor developments focus on its carbohydrate-binding pocket, which is necessary for its lectin activity in the extracellular space[33,34]. Inhibitors that would target the nanocluster enhancing function of Gal1 are missing.

Here we describe the identification of a 23-residue peptide that interferes with the binding of Gal1 to the RBD of Raf and disrupts H-Ras nanoclustering. Interestingly, this peptide broadly engages with a number of other RBD- and RA-domain containing Ras effectors, modulates Ras signaling and decreases cell viability.

## Results
### Galectin-1 binds via the RBD preferentially to B-Raf and stabilizes H-RasG12V nanoclustering

We previously provided evidence that Gal1, which can dimerize at higher concentrations, binds to the Ras binding domain (RBD) of Raf proteins to stabilize active H-Ras nanocluster[29] (Fig. 1a). We first corroborated some features of this stacked-dimer model using Bioluminesence Resonance Energy Transfer (BRET)-experiments. To this end, interaction partners

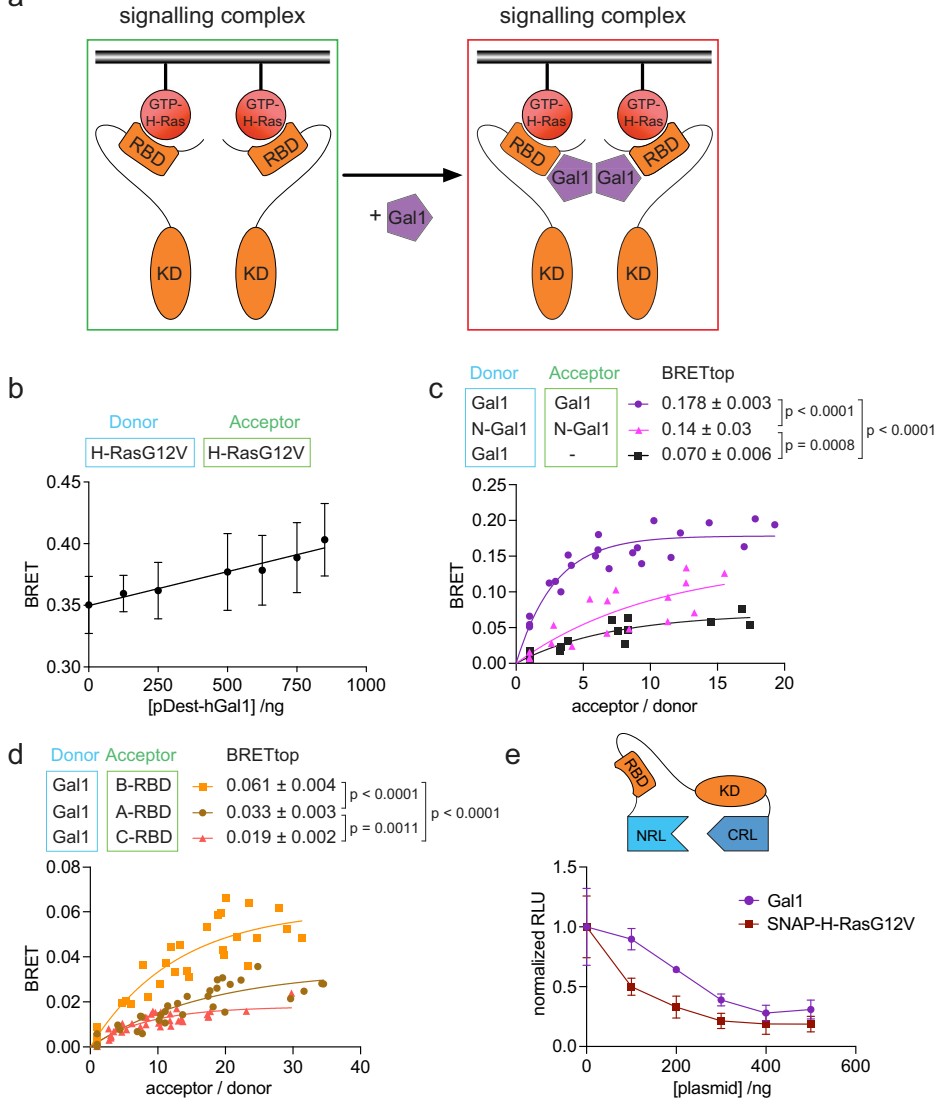

**Fig. 1 | The B-Raf preference of the H-Ras nanocluster scaffold Gal1 emerges within the RBD. a** Schematic of our model for Gal1 stabilized H-Ras nanocluster. **b** Dose-dependent effect of human Gal1 expression (48 h) on H-RasG12V nanoclustering-BRET (donor: acceptor plasmid ratio = 1:5); $n = 4$. **c** BRET-titration curves of the Gal1/ Gal1-interaction as compared to that of dimer-interface mutated N-Gal1. RLuc8-Gal1 was titrated with GFP2 as a control (black); $n = 3$. **d** BRET-titration curves of the Gal1-interaction with the RBDs of A-, B-, and C-Raf; $n = 3$. **e** Split-luciferase KinCon B-Raf biosensor response after expression of SNAP-H-RasG12V or Gal1; $n = 3$.

were tagged with RLuc8 or NanoLuc as donor and GFP2 or mNeonGreen as acceptor, and constructs were transiently expressed in HEK293-EBNA (hereafter HEK) cells to monitor the interaction by the increased BRET-signal. In BRET-titration experiments, the characteristic BRET-parameter BRETmax is typically determined. It is a measure for the maximal number of binding sites and the interaction strength, if other interaction parameters, such as complex geometry, are constant[35]. However, actual binding saturation is typically not reached in cells, and therefore BRETmax cannot be faithfully determined. Hence, we introduced the BRETtop value, which is the maximal BRET-ratio that is reached within a defined range of acceptor/donor signal-ratios, which is kept constant for BRET-pairs that are being compared[36].

In agreement with our earlier results obtained via Förster/fluorescence resonance energy transfer (FRET)[29], Gal1 expression increased H-RasG12V nanoclustering-BRET in a dose-dependent manner (Fig. 1b). Mutating four residues at the Gal1 dimer interface (N-Gal1) significantly reduced the BRETtop, suggesting that Gal1 is active as a dimer under our expression conditions[31] (Fig. 1c). BRET-experiments also confirmed the previously noted interaction preference of Gal1 for B-Raf[29] (Supplementary Fig. 1a), which was already seen with the RBDs of the corresponding Raf paralogs (Fig. 1d). Using computational docking that was based on experimentally determined constraints, we previously proposed a structural model for the binding of Gal1 to the RBD of C-Raf (C-RBD)[29] (Supplementary Fig. 1b). This model was validated by demonstrating that D113A, D117A mutations in the C-RBD significantly reduced binding to Gal1[29]. To further confirm these structural data, we here introduced analogous charge-neutralizing mutations D211A and D213A in the B-Raf-derived RBD (B-RBD), and mutation D75A in the A-Raf-derived RBD (A-RBD) (Supplementary Fig. 1c). In support of our docking data, the BRETtop of the interaction between Gal1 and either mutant was significantly reduced (Supplementary Fig. 1d, e). Consistent with the Raf-paralog specific interaction preference of Gal1, the mutated residues reside in a stretch that is least conserved between the RBDs (Supplementary Fig. 1c), which is in agreement with the significant difference in their Gal-1 BRET-interaction data (Fig. 1d).

Split-luciferase KinCon Raf-biosensors can report on the effect of mutations and modulators on the conformational state of Raf proteins[37] (Fig. 1e). The expression of SNAP-tagged oncogenic H-RasG12V (SNAP-H-RasG12V) reduces the luminescence signal, consistent with a relief of the closed autoinhibited state (Fig. 1e). Expression of increasing amounts of Gal1 likewise reduced the luminescence signal, suggesting that Gal1 facilitates the open state of B-Raf, although less than H-RasG12V (Fig. 1e).

Taken together with our previously published results[29], these data suggest a model wherein Gal1 binds to the RBD of Raf proteins, notably B-Raf, thus potentially destabilizing their autoinhibition. This could facilitate dimeric Ras–Raf engagement, which however requires a number of other modifications and conformational rearrangements[17]. When present as a dimer, Gal1 may further stabilize the active H-Ras/Raf stacked-dimer complex and thus an active H-Ras nanocluster, similar to what was observed with ON-state inhibitors of Raf[16].

### Identification of the L5UR-peptide as a disruptor of the Raf-RBD/galectin-1 interface

Gal1 increases H-Ras-driven MAPK output, and its elevated expression correlates with poorer survival in *HRAS* mutant cancers, such as head and neck squamous cell carcinoma, which frequently displays elevated Gal1 levels[22,29] (Supplementary Fig. 2a). Taken together with our H-Ras nanocluster model, these data support targeting of the interface between Gal1 and the Raf-RBD as a new strategy against oncogenic H-Ras. We hypothesized that the 52-mer L5UR peptide, which was derived from a Gal1 interaction partner, could act as a Raf-RBD/Gal1-interface inhibitor. Its residues 22–45 were previously shown to bind with a low affinity ($K_D = 310 \, \mu M$) to the opposite side of the carbohydrate binding site of Gal1[38]. This back-site overlaps with the one we had predicted as RBD-binding site on Gal1[29]. We thus expected that the L5UR-peptide would

disrupt the Raf-RBD/Gal1-interaction and consequently the Gal1-augmented H-RasG12V-nanoclustering and MAPK-signaling.

In line with this, expression of untagged L5UR decreased the FRET between mGFP-Gal1 and mRFP-C-RBD in HEK cells (Fig. 2a). This effect was comparable to the loss observed in the C-RBD-D117A mutant with reduced Gal1-binding (Fig. 2a)[29]. For comparison, we tested the effect of Anginex and its topomimetic small molecule analog OTX-008[39]. Anginex is a 33-mer angiostatic peptide that binds to Gal1 at an unknown site[40–42]. Competitive fluorescence polarization experiments with FITC-tagged full-length L5UR (F-L5UR) as a probe, established that it can be displaced from purified His-tagged Gal1 by the Anginex peptide (Supplementary Fig. 2b). However, neither Anginex nor OTX-008 disrupted the Gal1/C-RBD interaction as measured by FRET in cells (Fig. 2a), suggesting that the Anginex binding site only partially overlaps with the L5UR-binding site, but not sufficiently with the C-RBD binding site on Gal1. By contrast, expression of the L5UR-peptide decreased the Gal1-augmented H-RasG12V nanoclustering-FRET (Fig. 2b). In agreement with previous data[29], dimerization-deficient N-Gal1 did not increase nanoclustering-FRET, and co-expression of the L5UR-peptide had no additional effect (Fig. 2b).

Next, we aimed to confirm that L5UR engages directly with the Raf-RBD/Gal1 interface. We purified His-tagged Gal1 and the GST-tagged B-RBD and performed pulldown experiments with a biotin-tagged L5UR (bio-L5UR) peptide (Fig. 2c). Interestingly, L5UR pulled down Gal1 and the GST-B-RBD independently from each other (Fig. 2c). Indeed, fluorescence polarization binding experiments determined a micromolar ($K_D = 7.3 \pm 0.7 \, \mu M$) binding of F-L5UR to the GST-B-RBD (Fig. 2d), but no binding to GST alone (Supplementary Fig. 2c). Using a Quenching Resonance Energy Transfer (QRET)-assay, we independently confirmed the micromolar binding to B-RBD, even with the shortened 22–44 residue core fragment of L5UR labeled with a europium-chelate (Eu-L5URcore) (Table 1, Supplementary Fig. 2d). By contrast, no saturation of Eu-L5URcore binding to Gal1 could be observed at the technically highest possible concentration of 135 $\mu M$ (Supplementary Fig. 2e).

Competitive fluorescence polarization experiments, using F-L5UR as a probe, established that the full-length peptide of L5UR could be displaced from the GST-B-RBD with an $IC_{50} = 2 \pm 1 \, \mu M$ (Fig. 2e), and likewise from the C-RBD (Table 1, Supplementary Fig. 2f). As expected, the shorter L5URcore could displace F-L5UR from the C-RBD with a slightly reduced potency ($IC_{50} = 14 \pm 6 \, \mu M$) (Supplementary Fig. 2f). The L5UR has a high proportion of six positively charged arginine residues in its core region, which may indicate that binding of the peptide to the RBD of Raf is influenced by electrostatic interactions. We therefore introduced several negatively charged, acidic residues to mostly replace basic and hydrophobic residues in the core-region of the L5UR peptide to generate a non-binding mutant (mutL5UR) (Fig. 2f). Indeed, mutL5UR did not have any displacement activity in the competitive fluorescence polarization assay (Fig. 2e, Supplementary Fig. 2f). Circular dichroism spectra of the L5UR, L5URcore and mutL5URcore peptides suggested they were mostly random coil with ~25% antiparallel β-sheet (Supplementary Fig. 2g).

In conclusion, L5UR binds with low micromolar affinity to the RBDs of B-Raf and C-Raf (Table 1). This interaction is significantly determined by residues in its core region, as binding is attenuated in the mutL5UR variant.

### SNAP-tagged L5UR disrupts the B-RBD/ galectin-1 complex, and H-RasG12V nanoclustering in cells and binds to multiple Ras interactors

To improve the readout of L5UR-variant expression in cells and eventually enable further functionalization, we designed genetic constructs where a SNAP-tag was added via a long linker to the C-terminus of the peptide (Fig. 3a).

The L5UR-SNAP dose-dependently decreased BRET between Gal1 and the B-RBD to a similar extent as the untagged L5UR, confirming that the SNAP-tag did not increase activity further (Fig. 3b). In agreement with the binding data (Fig. 2e), mutL5UR-SNAP did not decrease the BRET signal, nor did the SNAP-tag alone (Fig. 3b). Immunoblotting confirmed an

**Fig. 2 | The L5UR-peptide binds to the Raf-RBD and disrupts the Raf-RBD/ Gal1-complex. a** Effect of L5UR expression (24 h) on Gal1/C-RBD FRET (donor:acceptor plasmid ratio = 1:3); $n$ = 3. **b** Effect of L5UR expression (24 h) on Gal1-augmented H-RasG12V nanoclustering-FRET (donor:acceptor plasmid ratio = 1:3); $n$ = 3. **c** Immunoblot data from pull-down assay with biotinylated L5UR and purified Gal1, GST-B-RBD or GST-only control with example blots (left) and quantification of repeat data (right); $n$ = 3. **d** Binding of 10 nM F-L5UR to GST-B-RBD detected in a fluorescence polarization assay; $n$ = 3. **e** Displacement of F-L5UR (10 nM) from GST-B-RBD (15 μM) by L5UR-derived peptides; $n$ = 3. **f** Sequences of L5UR-derived peptides as used for in vitro and in cellulo assays. The stretch of the core peptide is highlighted in blue, mutations are in red.

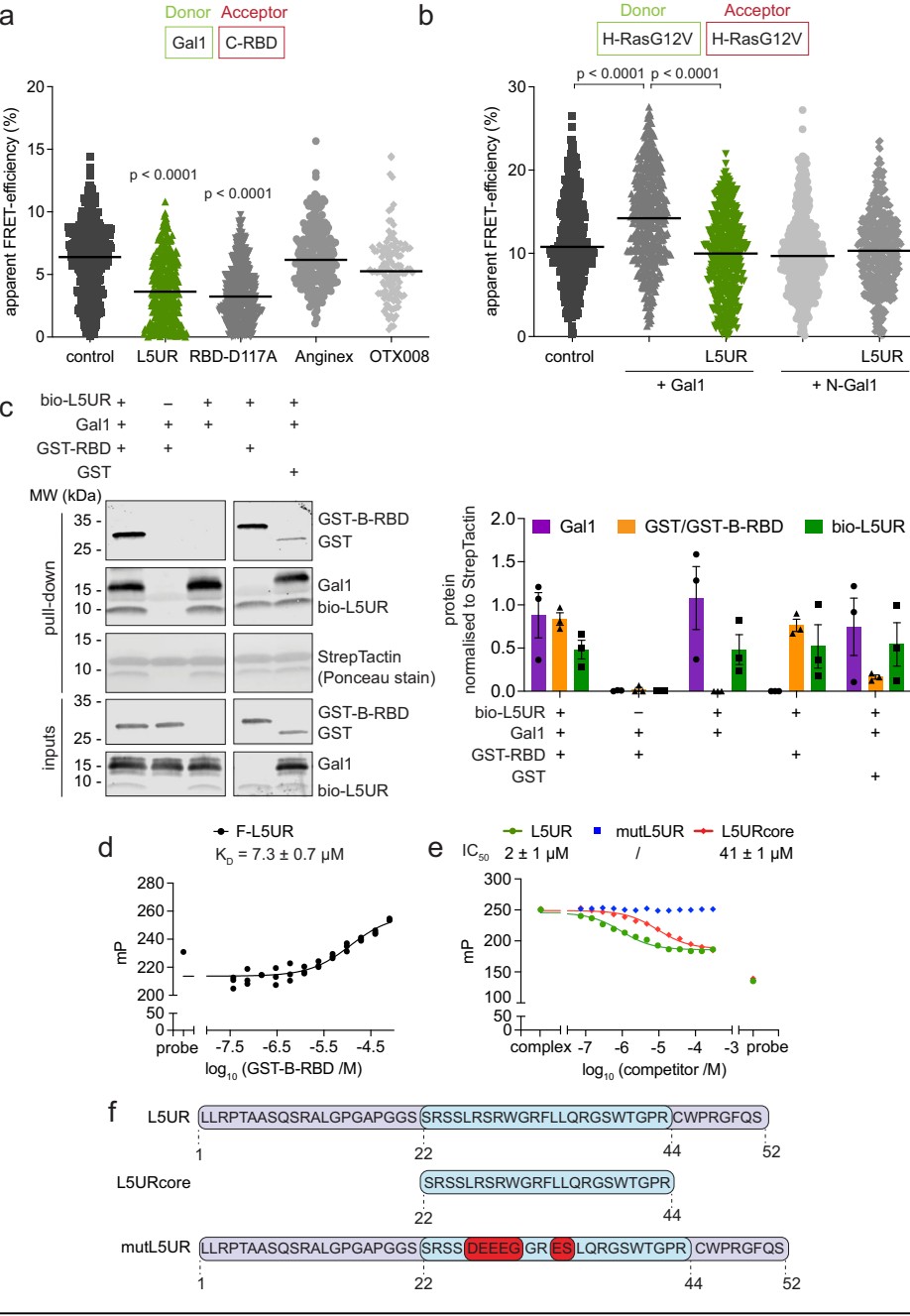

## Table 1 | Overview of L5UR/ Raf-RBD in vitro binding data

| Protein | Probe | $K_D$[b] or IC$_{50}$/μM |
|---|---|---|
| GST-B-RBD | L5UR | 7.3 ± 0.7[b] |
| GST-B-RBD | L5UR[a] | 2 ± 1 |
| GST-B-RBD | L5URcore[a] | 41 ± 1 |
| B-RBD | L5URcore (QRET) | 18 ± 1 |
| C-RBD | L5UR[a] | 4 ± 1 |
| C-RBD | L5URcore[a] | 14 ± 6 |

[a]In competitive fluorescence polarization assay with F-L5UR.
[b]Marks actual $K_D$, while otherwise IC$_{50}$ are reported.

initially linear increase of L5UR-SNAP variant expression with increasing amounts of transfected constructs (Supplementary Fig. 3a, b). Consistent with the Gal1/ B-RBD disruption, the L5UR-SNAP construct decreased Gal1-enhanced H-RasG12V nanoclustering-BRET to a similar extent as the untagged L5UR, while again mutL5UR or the SNAP-tag alone had no effect (Fig. 3c). In line with the higher affinity of L5UR for the Raf-RBD, we observed very similar effects even without co-expression of Gal1 in HEK cells that are otherwise comparatively devoid of Gal1 (Supplementary Fig. 3c, d). L5UR or L5UR-SNAP reduced the nanoclustering-BRET by ~33% (Fig. 3c), while co-expression of SNAP-H-RasG12V led to a ~85% reduction (Supplementary Fig. 3e). Neither of the L5UR-constructs significantly perturbed K-RasG12V nanoclustering-BRET, suggesting a potential Ras isoform selectivity (Supplementary Fig. 3f).

The disruption of H-RasG12V nanoclustering specifically by L5UR-SNAP, but not the SNAP-tag alone, was furthermore confirmed by the classical electron microscopy-based Ras nanoclustering analysis performed

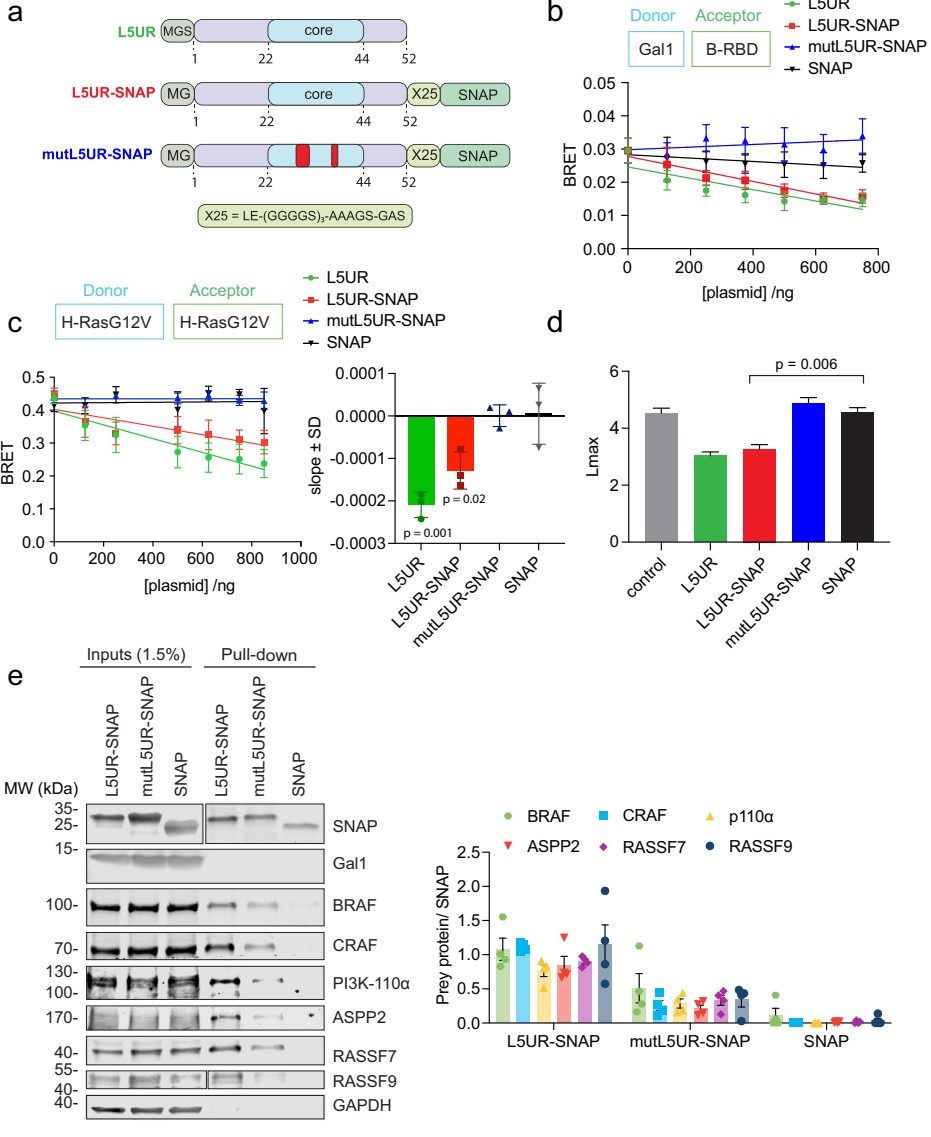

**Fig. 3 | The L5UR and L5UR-SNAP peptides disrupt H-RasG12V nanoclustering. a** Schematics of L5UR derived constructs expressed in cellular assays. The stretch of the core peptide is highlighted in blue, loss-of-function mutations are indicated red. **b** Effect of expression of L5UR constructs (48 h) on Gal1/B-RBD BRET (donor:acceptor plasmid ratio = 1:10); $n$ = 3. **c** Effect of L5UR construct expression (48 h) on H-RasG12V nanoclustering-BRET with co-transfection of 200 ng Gal1 plasmid (donor:acceptor plasmid ratio = 1:5); $n$ = 3. Statistical comparison was done against the SNAP-only sample. **d** Electron microscopy-based analysis of H-RasG12V nanoclustering in BHK cells showing the effects of L5UR-construct expression and controls; $n$ = 15. Higher Lmax values indicate higher nanoclustering. **e** Immunoblot data from pull-down assays with L5UR-SNAP and control constructs from HEK cells co-expressing Gal1 with example blots (left) and quantification of repeat data (right); n = 4 (left).

on membrane sheets of Gal1-expressing BHK cells (Fig. 3d)[21]. These data therefore confirmed the disruption of H-RasG12V nanoclustering by L5UR- and L5UR-SNAP construct expression.

While Gal1 appears to have a preference for B-Raf, it readily engages with the RBD of other Raf proteins (Fig. 1d, Supplementary Fig. 1a). We therefore tested if L5UR can also bind to other RBD- and RA-containing proteins by performing pull-down experiments. The SNAP-tag enabled covalent coupling of L5UR or mutL5UR to beads that were incubated with lysates of Gal1-transfected HEK cells. While the SNAP-tag alone did not interact with any of the examined proteins (Fig. 3e), L5UR-SNAP pulled down not only full-length B-Raf and C-Raf, but also the catalytic subunit of PI3Kα. ASPP2 contains an RBD, interacts with oncogenic H-Ras and is a pan-Ras nanocluster scaffold that can neutralize Gal1 nanoclustering and can switch from a Gal1 promoted growth to a senescence phenotype[43–45]. Like the other RBD-containing proteins it was pulled down by L5UR-SNAP, as were its two RA-domain containing interaction partners, RASSF7 and RASSF9, which do not directly bind to Ras[5,46]. Quantification confirmed that the mutL5UR-SNAP was ≤50% more efficient than L5UR-SNAP in pulling down any of these proteins (Fig. 3e). It is therefore likely that downstream of Ras and other small GTPases several pathways are affected by L5UR.

## TAT-tagged L5UR modulates Ras-signaling and weakly inhibits cell proliferation

Peptides can be rendered cell-permeable by the addition of cell penetrating sequences, which facilitate their characterization as prototypic and proof-of-concept reagents[47]. The 12-residue cell penetrating TAT-peptide that is derived from a Human Immunodeficiency Virus (HIV)-protein, can facilitate cellular peptide uptake[48–50]. We therefore chemosynthetically added the TAT-peptide via a PEG2-linker to the 23-residue long L5URcore peptide (TAT-L5URcore) and the corresponding loss-of-function mutant (TAT-mutL5URcore) (Fig. 4a).

To verify cell penetration and on-target activity, we tested the effect of the TAT-peptides in our on-target BRET-assays. Both the BRET between Gal1 and the B-RBD (Fig. 4b), as well as H-RasG12V-nanoclustering BRET (Fig. 4c), were dose-dependently decreased by the TAT-L5URcore peptide with $EC_{50}$ = 16 ± 1 μM and $EC_{50}$ = 19 ± 1 μM, respectively. Neither the TAT-peptide alone, nor the mutant TAT-mutL5URcore, or the non-TAT peptides L5URcore and mutL5URcore decreased the BRET-signal in either assay (Fig. 4b, c).

Based on our model and mechanistic data, signaling, and proliferation of *HRAS* mutant cancer cell lines with high Gal1 levels were expected to respond best to the nanocluster disrupting TAT-L5URcore peptide. Cancer

**Fig. 4 | The TAT-tagged L5URcore peptide disrupts H-RasG12V nanoclustering. a** Schematics of TAT-functionalized L5URcore-derived peptides and controls as applied in cellular assays. Loss-of-function mutations of L5UR are indicated in red. Non-TAT peptides are acetylated at the N-terminus. **b**, **c** Effect of cell-penetrating derivatives of L5URcore and control peptides on Gal1/B-RBD BRET (**b** donor:acceptor plasmid ratio = 1:10; $n \geq 2$) or H-RasG12V nanoclustering-BRET (**c** donor:acceptor plasmid ratio = 1:5, co-transfection of 200 ng Gal1 plasmid; $n = 3$). After 24 h expression of plasmids, peptides were added to cells at specified concentrations and incubated for 2 h.

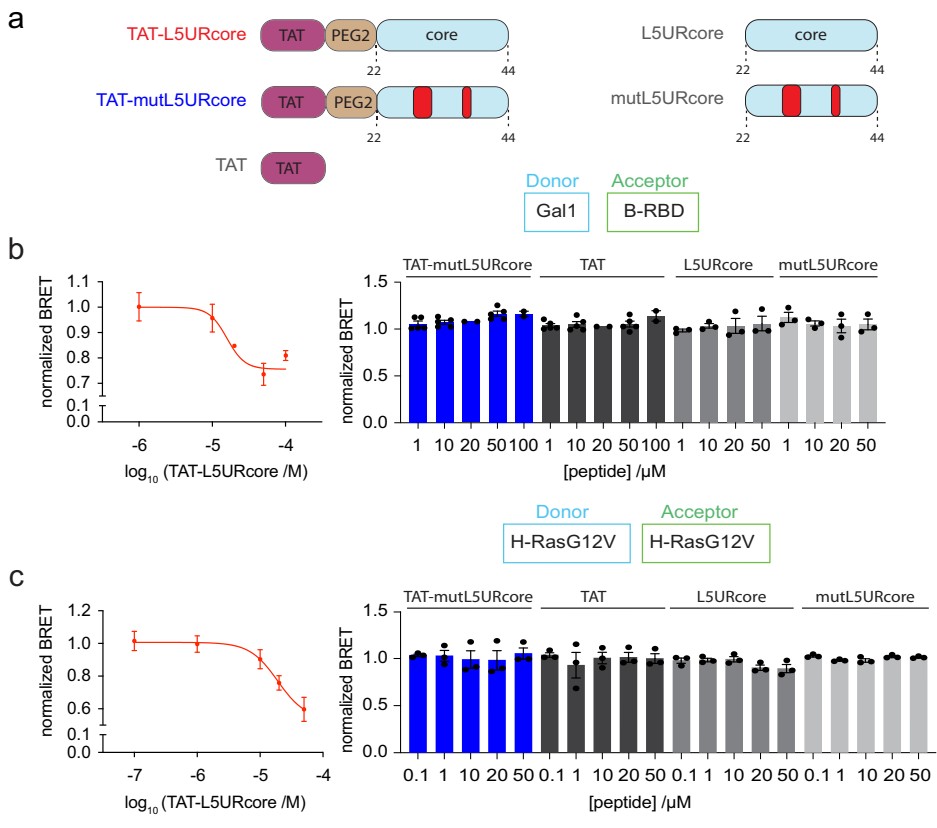

cell lines Hs 578 T (*HRAS-G12D*) and T24 (*HRAS-G12V*), as well as the *KRAS-G12C* mutant MIA PaCa-2, express high levels of Gal1, while HEK cells have, in comparison undetectably low levels of Gal1 (Supplementary Fig. 3d).

Indeed, treatment of the *HRAS*-mutant cell lines Hs 578 T (Fig. 5a, e) and T24 (Fig. 5b, f) specifically with the TAT-L5URcore peptide at 1–20 µM reduced EGF-induced cellular pERK-and pAkt-levels in a dose-dependent manner. In MIA PaCa-2 pERK remained unaffected, while pAkt-levels were reduced at 20 µM (Fig. 5c, g). Interestingly, in non-transformed HEK cells pERK-levels were slightly induced by TAT-L5URcore, as were pAkt-levels, which were however also upregulated by trametinib (Fig. 5d, h). Furthermore, an apparently non-specific increase in pERK- and/or pAkt-levels was observed at intermediate concentrations of TAT-mutL5URcore and TAT notably in Hs 578 T and MIA PaCa-2. TAT-L5URcore effects on signaling were still relatively weak, which can be attributed to the immaturity of this reagent with only micromolar activity.

We then examined the effect of the TAT-enabled peptides on the viability of these cell lines. The proliferation of *HRAS*-mutant cancer cell lines Hs 578 T (Fig. 6a, e) and T24 (Fig. 6b, e) was specifically reduced by TAT-L5URcore, but not the control TAT-peptides. However, this was also observed for *KRAS*-mutant MIA PaCa-2 (Fig. 6c, e) and non-transformed HEK cells (Fig. 6d, e). To quantitate the relatively weak effect of the peptides on cell proliferation more accurately, we applied the normalized area under the curve DSS3-analysis, where a higher DSS3-score corresponds to higher anti-proliferative activity (Fig. 6e). While we observed a higher anti-proliferative effect of TAT-L5URcore as compared to TAT-mutL5URcore, the broad effect on cell proliferation may indicate that the TAT-L5URcore interferes with several signaling pathways that are relevant for cell proliferation and survival.

## Discussion

We here demonstrate that the 23-residue L5URcore peptide binds with micromolar affinity to the Raf-RBD at a site that enables it to disrupt the interaction with Gal1. The peptide interferes with nanocluster of active H-Ras and inhibits Ras-signaling and cell proliferation. The fact that L5UR reduces nanoclustering of H-Ras even in HEK cells that have very low Gal1 levels, is consistent with its higher affinity to Raf-RBDs than to Gal1. Yet, it is plausible that by interfering at the Raf-RBD/ Gal1 interface, L5UR can unfold a higher and more selective activity in *HRAS*-mutant cells with high Gal1 levels, such as observed for Hs 578 T and T24 (Fig. 5). However, the broad impact on cell proliferation (Fig. 6), its engagement of several Ras interactors (Fig. 3e), and its mixed effect on signaling (Fig. 5), suggest that L5URcore is still an immature tool reagent. It nevertheless represents a starting point for the development of novel Ras-nanocluster disrupting reagents that engage with one or more Ras-interactors to affect Ras-signaling and cancer cell proliferation.

How selectively L5UR disrupts the H-Ras nanocluster remains unclear. It is currently unknown how Gal1 positively regulates H-Ras nanocluster but negatively K-Ras nanocluster[29]. Vice versa, how the related galectin-3 (Gal3) increases, specifically K-Ras nanocluster is not known[51–53]. In the context of our stacked-dimer model (Fig. 1a) and our KinCon-data (Fig. 1e), it is conceivable that galectins facilitate the activation of Raf and/or stabilize specific Raf-dimers to facilitate nanoclustering of specific Ras isoforms. Indeed, Gal1 distinguishes between the RBDs from A-, B-, and C-Raf and most strongly engages the B-Raf-RBD. For K-Ras, evidence exists that it binds preferentially with B-/C-Raf-dimers[16,54], while for Gal1 augmented H-Ras nanocluster our previous data suggested a particular relevance for B-/A-Raf dimers[29]. One would, therefore, predict that these dimers are specifically stabilized by Gal3 and Gal1, respectively. However, it is not entirely plausible how symmetrical dimers of galectins, or in the case of Gal3 potentially even oligomers[25], would stabilize asymmetric dimers of Raf proteins. Heterodimerization of galectins could provide a solution to this problem. In humans, 15 different galectins are found and only Gal1 and Gal3 are characterized as nanocluster scaffolds so far[25]. Given the relatedness in this protein family, it is plausible to assume that other galectins have a similar activity and potentially mixed galectin-dimers could form that then

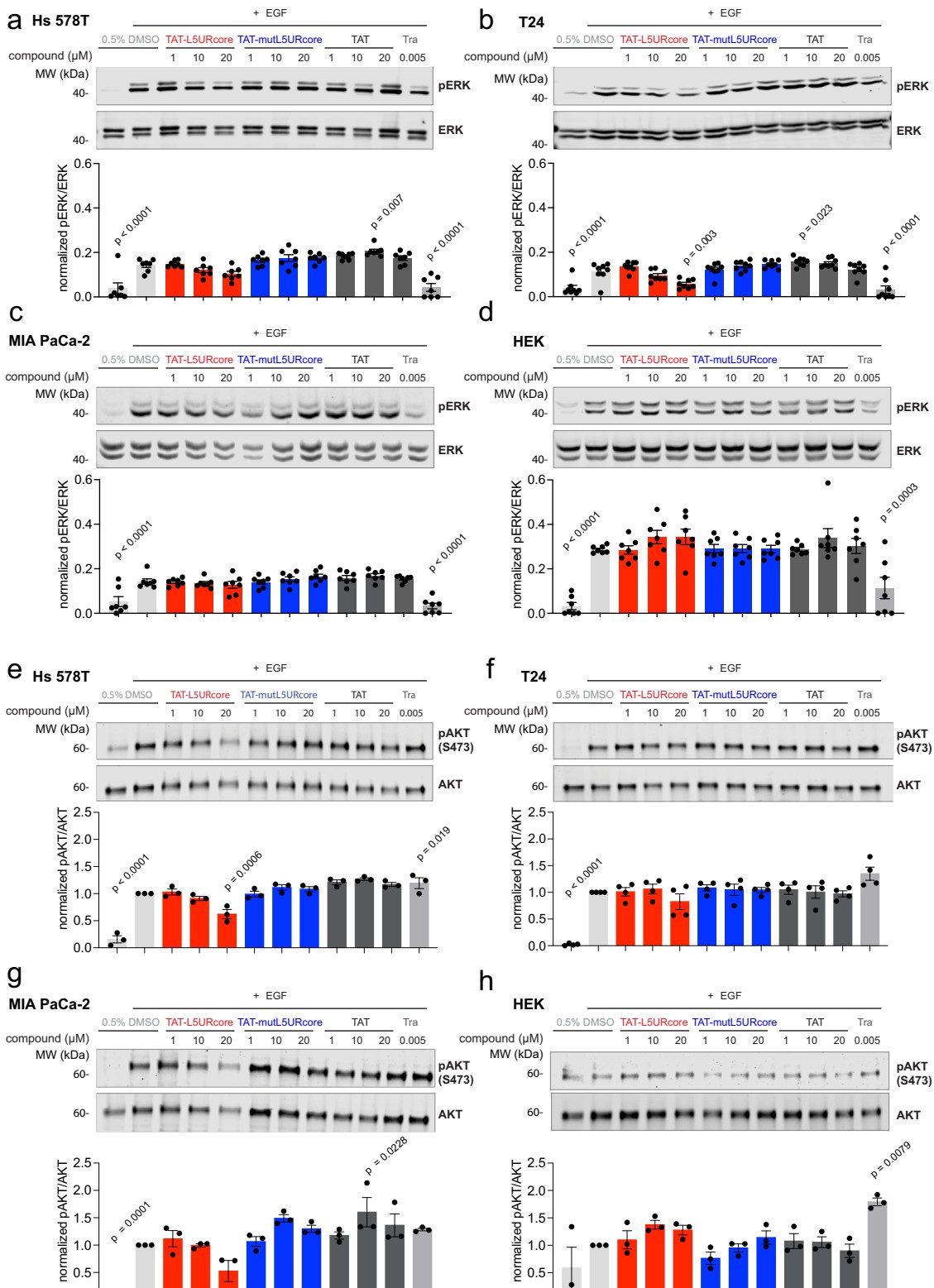

**Fig. 5 | The TAT-tagged L5URcore peptide impacts on Ras-signaling. a–h** Immunoblot analysis of lysates from Hs 578 T (**a**, **e**), T24 (**b**, **f**), MIA PaCa-2 (**c**, **g**), and HEK (**d**, **h**) cells after EGF-stimulation and treatment with L5URcore-derived peptides with and without TAT-tag or control compound, trametinib (Tra), for 2 h; $n$ = 3–8.

stabilize the asymmetric dimers of Raf. Therefore, a complex equilibrium of mixed oligomers that partly stabilize and partly compete and sequester could be the answer to the intricate problem of Ras-isoform specific nanoclustering effect of galectins.

The TAT-L5URcore peptide provides a unique tool to investigate the functioning of Ras nanocluster further. In contrast to current galectin inhibitors, which target the carbohydrate-binding pocket[33,34], the L5UR-peptide acts via a novel mode-of-action that at least in part exploits the role

**Fig. 6 | HRAS-mutant cancer cell proliferation is decreased by TAT-L5UR peptides. a–d** 2D cell viability of Hs 578 T (**a**), T24 (**b**), MIA PaCa-2 (**c**), and HEK (**d**) cells in response to 48 h treatment with TAT-L5URcore peptides and TAT-control; $n = 3$. **e** Drug sensitivity score (DSS3), an area under the curve metric, calculated for the viability data in (**a–d**). A higher value indicates a stronger anti-proliferative effect. TAT-control was used as a reference for statistical comparisons.

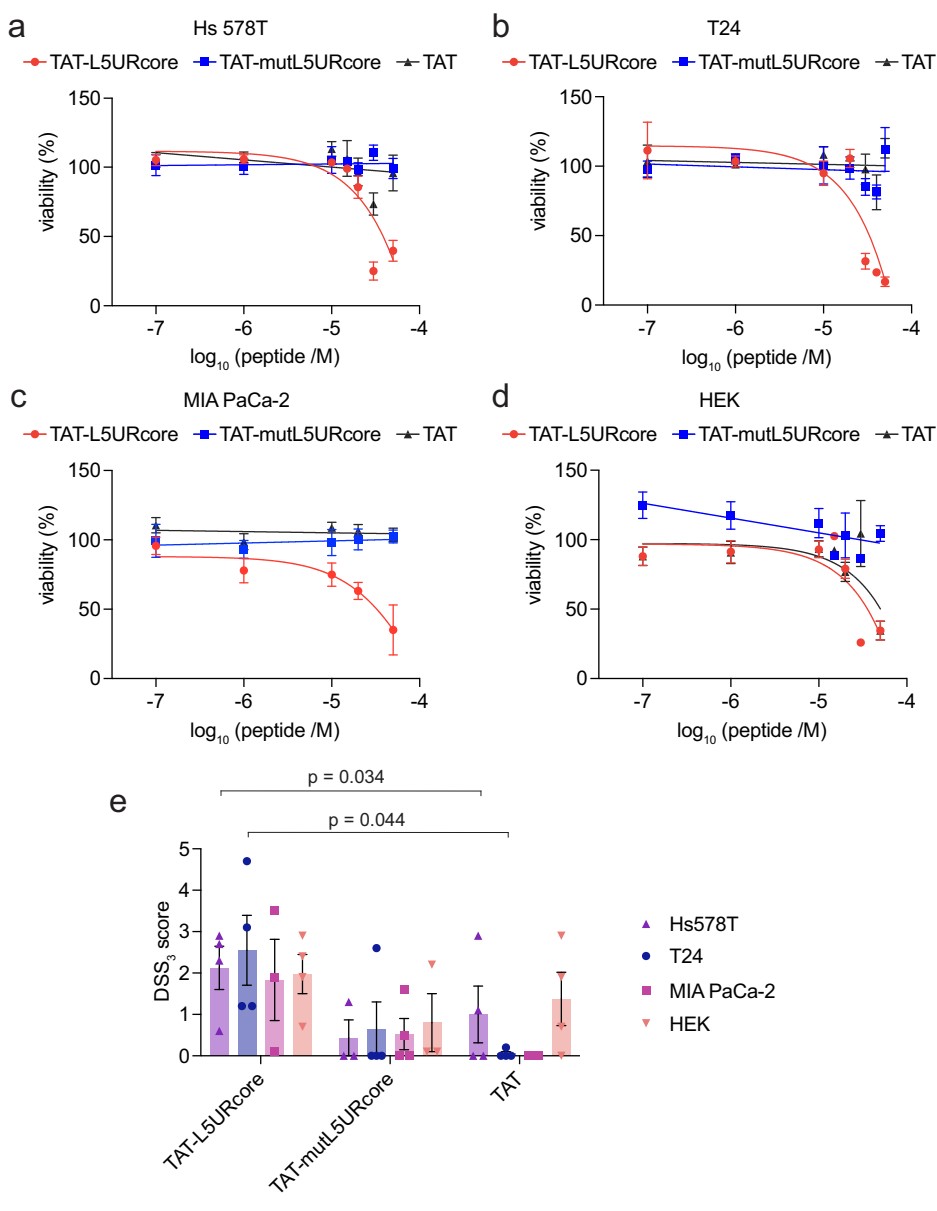

of the Raf-RBD/ Gal1 interface in nanocluster stabilization. The intermediate size below 3 kDa of the TAT-L5URcore peptide represents a relevant starting point for the development of smaller molecules with analogous mode-of-action[55,56]. The properties of this peptide and the putative target site suggest that not a distinct pocket, but an assembly of charged and hydrophobic interactions are the major driving force for its affinity. Regarding size, mechanism-of-action, and specificity, L5URcore contrasts with the NS1-monobody, which specifically binds to the allosteric lobe of K-Ras and H-Ras to disrupt nanoclustering[57]. Given the size of the monobody of ~10 kDa it is likely that the steric hindrance caused by this large ligand is mostly responsible for the interference with nanoclustering. With the identification of the targetable site on the Raf-RBD and with more insight into the structure of the Gal1/RBD complex, it will be possible to identify improved binders with higher affinity and specificity in the future. Both competitive screening as well as the structure-based design of peptidomimetics present opportunities for future improvements. The fact that multiple RBD- and RA-domain proteins are bound by L5UR may in this context at first appear as a liability but may hold the opportunity to develop novel RBD- and RA-binders that could affect a broad range of effectors.

Targeting of the H-Ras nanocluster scaffold Gal1 is quite different from approaches focusing on the main nodes of the Ras-MAPK pathway. Both mechanistic and genetic evidence suggest that Gal1 acts as a positive modifier that is associated with a worse progression of *HRAS* mutant cancers, notably head and neck cancers that are frequently associated with high Gal1 levels (Supplementary Fig. 2a). While *HRAS* is overall the least frequently mutated *RAS* gene (in 1.3% of cancer patients), it is mutated in >5% of head and neck squamous cell carcinomas (HNSC)[58]. The prognosis for patients with recurrent and metastatic HNSC is still poor[59]. While tipifarnib, a farnesyltransferase inhibitor shows promising efficacy in HNSC patients, there is still a need for potent treatments[60]. By interfering at the interface of Raf-proteins with Gal1, one may not eliminate other functions of Gal1 and modulate Raf in an unconventional manner that may allow for a normalization of the signaling activity. This would be beneficial regarding side effects, as normal tissue functions could continue to progress.

We expect that our L5UR peptide work will provide new perspectives on how to target Ras nanocluster and potentially also several Ras interactors in a different way.

## Methods

### Expression constructs

Here we refer to the 52-mer fragment derived from residues 38–89 of the unique region of the λ5-chain (λ5-UR) of the pre-B-cell receptor as L5UR. This unique region bears no similarity to known proteins[38]. The pClontech-L5UR was made by excising L5UR cDNA from pET28a-L5UR (gift from Dr. Elantak), using NheI–XhoI sites and subcloned into pmCherrry-C1 (Clontech, #632524). This removed the mCherry cDNA from the expression vector leaving only the full-length L5UR. Vector pcDNA-Hygro-Anginex was a gift from Prof. Thijssen[42,61]. Expression clones were mostly produced by multi-site gateway cloning as described in our previous studies[36,62,63]. Some expression clone genes were synthesized and cloned into desired vectors by the company GeneCust, France. The B-Raf KinCon sensor encoded by the pcDNA3-RLucF1-BRAF-RLucF2 plasmid was described previously by others[37]. A listing of all plasmid constructs and their sources is provided in Supplementary Table 1.

### Cell culture

Hs 578 T, T24, MIA PaCa-2, and BHK-21 cells were obtained from DSMZ-German Collection of Microorganisms and Cell Cultures GmbH or ATCC. HEK293-EBNA cells were a gift from Prof. Florian M. Wurm, EPFL, Lausanne. All cell lines were cultured in a humidified incubator maintained at 37 °C and 5% $CO_2$, in Dulbecco's modified Eagle Medium (DMEM) (Gibco, #41965039) supplemented with 9% (v/v) Fetal Bovine Serum (FBS) (Gibco, #10270106), 2 mM L-Glutamine (Gibco, #25030081) and penicillin-streptomycin (Gibco, #15140122) 10,000 units/mL (complete growth medium), in T75 culture flasks (Greiner, #658175). Cells were regularly passaged 2–3 times a week and routinely tested for mycoplasma contamination using MycoAlert Plus mycoplasma Detection kit (Lonza, #LT07-710).

### Bacterial strains

Competent E. coli BL21 Star (DE3)pLysS and E. coli DH10B were grown in Luria–Bertani (LB) medium (Sigma, #L3022) at 37 °C, with appropriate antibiotics unless otherwise stated.

### Peptide synthesis

Reagents were purchased from Iris Biotech GmbH, Sigma Aldrich, and Carl Roth and used without additional purification. Synthetic protocols were adapted from previously reported protocols[64–66]. All reaction steps were performed in a syringe reactor at room temperature on an orbital shaker. Unless stated otherwise, all procedures were performed with 1 mL of solvent or reagent solution per 50 mg resin. For all scales (10–100 μmol), H-Rink amide ChemMatrix® resin (Sigma-Aldrich, Art. No. 727768) was swollen in dimethylformamide (DMF) for 30 min. For amino acid (aa) coupling a solution of 4 eq. N-α-fluorenylmethyloxycarbonyl (Fmoc) protected amino acid, 4 eq. (1-cyano-2-ethoxy-2-oxoethyliden-aminooxy)dimethylamino-morpholino-carbenium hexafluorophosphate (COMU) and 4 eq. ethyl cyano (hydroxyimino) acetate (Oxyma) in DMF was prepared (0.3 mL per 50 mg resin). Then 8 eq. N,N-diisopropylethylamine (DIPEA) were added to the coupling solution. Subsequently, resin and coupling solution were mixed in a syringe reactor on an orbital shaker. After 30 min, the reaction solution was discarded, and the amino acid coupling was repeated once. After washing the resin with DMF (3×), dichloromethane (DCM) (3×), and DMF (3×), Fmoc removal was performed by adding a solution of piperidine in DMF (2:8, v/v). After 5 min, the solution was discarded and the Fmoc removal was repeated. The resin was washed with DMF (3×), DCM (3×), and DMF (3×). Afterwards, subsequent amino acids were added by repeating cycles of amino acid coupling and Fmoc removal. Peptide synthesis was supported by automated solid-phase synthesis (SPPS), using the peptide synthesis robot Syro I (MultiSynTech), with a double coupling protocol of 4 eq. benzotriazol-1-yl-oxytripyrrolidinophosphonium hexafluorophosphate (PyBOP, 1st 40 min coupling,) and 4 eq. hexafluorophosphate azabenzotriazole tetramethyl uronium (HATU, 2nd 40 min coupling) as coupling reagents and DMF as solvent. Additionally, the

coupling reaction with 4 eq. The Fmoc-protected amino acid was supplemented with 4 eq. Oxyma and 8 eq. DIPEA. Before Fmoc removal was conducted with 25% (v/v) piperidine in DMF, a capping step using $Ac_2O$ (acetic anhydride) and DIPEA in NMP (1:1:8, v/v/v) was performed. In between reaction steps, the resin was washed with DMF.

N-terminal acetylated peptides were synthesized by adding a solution of acetic anhydride, DIPEA, and DMF (1:1:8, v/v/v) to the immobilized peptide on resin. The reaction solution was discarded after 10 min, and the acetylation was repeated. Subsequently, the resin was washed with DMF (3×), DCM (3×), and DMF (3×).

N-terminal biotin labeled peptides were prepared by coupling the linker 18-(9-Fluorenylmethyloxycarbonylamino)-4,7,10, 13-tetraoxa-octadecanoic acid (Fmoc-PEG5-OH) as described above onto the N-terminus. After Fmoc removal (see above), biotin was coupled as described above but increasing to 6 eq. of biotin. Subsequently, the resin was washed DMF (3×), DCM (3×) and DMF (3×).

Peptide cleavage and removal of side chain protecting groups were performed simultaneously by adding a solution of trifluoroacetic acid (TFA), triisopropylsilane (TIPS), 1,8-octanedithiol (ODT), and water (94:2.5:2.5:1, v/v/v/v) to the resin. After 2 h, the cleavage solution was collected and evaporated. The crude peptides were obtained by precipitation in diethyl ether with subsequent centrifugation (10 min, 4000 rcf). After the removal of the supernatant, the crude peptide was dissolved in acetonitrile (ACN) and water (1:4, v/v).

Peptide purification was carried out via reverse-phase HPLC (high-performance liquid chromatography) on an Agilent semi-preparative system 1100 (Column: Macherey-Nagel Nucleodur C18, 10 × 125 mm, 110 Å, 5 μm) using various gradients of solvent A ($H_2O$ + 0.1% TFA) and solvent B (ACN + 0.1% TFA) over 20–40 min with a flow rate of 6 mL min$^{-1}$.

Peptides were analyzed by analytical reverse-phase HPLC coupled to ESI-MS (Agilent 1260 + quadrupole 6120, Column: Eclipse XDB-C18, 4.6 × 150 mm, 5 μm) with solvent A ($H_2O$ + 0.1% FA + 0.01% TFA) and solvent B (ACN + 0.1% FA + 0.01% TFA) via a 10 min gradient from 5% to 95% solvent B. An overview of peptides synthesized by us in this study is given in Supplementary Table 2.

### Protein purification

For protein expression, a 16 h culture was set by inoculating colonies into an appropriate volume of antibiotic-supplemented LB media incubated 16 h at 37 °C. The next day, 25 mL of the culture was added to 1 L of LB and incubated at 37 °C until OD at 600 nm reached 0.6–0.9, at which point protein expression was induced by adding isopropyl $\beta$-D-1-thiogalactopyranoside (IPTG) (VWR, #437145X) at the final concentration of 0.5 mM. GST-tagged B-Raf-RBD (residues 155–227 of human B-Raf) and GST-tagged C-Raf-RBD (residues 50–134 of human C-Raf) protein expression was induced for 4 h at 23 °C, and the His-tagged protein expression was induced for 16 h at 25 °C. Afterward, the cell pellet was collected by centrifugation, rinsed in PBS, and stored at −20 °C until purification.

For GST-tagged protein purification, cells were lysed by resuspending the pellet in a buffer consisting of 50 mM Tris-HCl pH 7.5, 150 mM NaCl, 2 mM DTT, 0.5% (v/v) Triton-X 100, 1× Protease Inhibitor Cocktail (Thermo Scientific Pierce Protease Inhibitor Mini Tablets, EDTA-free, #A32955) and by sonication on ice using a Bioblock Scientific Ultrasonic Processor instrument (Elmasonic S 40 H, Elma). Lysates were cleared by centrifugation at ~18,500×g for 30 min at 4 °C. For GST-tagged proteins, the cleared lysate was incubated with 500 μL glutathione agarose slurry (GE Healthcare, #17-0756-01) (resuspended 1:1 in lysis buffer) for 3 h at 4 °C with gentle rotation. Next, the supernatant was removed, and beads were washed five times with 1 mL of washing buffer consisting of 50 mM Tris-HCl at pH 7.5, 500 mM NaCl, 5 mM DTT, 0.5% (v/v) Triton-X 100. Next, beads were rinsed three times with 1 mL of equilibration buffer (50 mM Tris-HCl pH 7.5, 150 mM NaCl, 2 mM DTT). GST-tagged protein was eluted off the beads by using a 20 mM glutathione solution (Sigma-Aldrich, #G4251-5G). Fractions were analyzed by resolving on 4–20% gradient SDS-

PAGE (BioRAD #4561094 or #4651093), stained with Roti Blue (Carl Roth Roti-Blue quick, #4829-2), and dialyzed into a final dialysis buffer (50 mM Tris-HCl at pH 7.5, 150 mM NaCl, 2 mM DTT, 10% (v/v) glycerol) by using a D-Tube Dialyzer with MWCO 6–8 kDa (Millipore, #71507-M) for 16 h at 4 °C. Protein concentration was measured using NanoDrop 2000c Spectrophotometer (Thermo Fischer Scientific) and stored at −80 °C.

For GST-tag removal, the cleared lysate was incubated with 500 μL of glutathione agarose slurry (resuspended 1:1 in lysis buffer) for 5 h at 4 °C with gentle rotation, then proceeded to washing steps as described above. The beads were rinsed with equilibration buffer and then with dialysis buffer before the excess was drained as much as possible. The beads were then resuspended in 650 μL of dialysis buffer and 100 U of Thrombin (GE Healthcare, #GE27-0846-01), to a final volume of 1 mL. The next day, the untagged protein was collected by applying supernatant to 1 mL polypropylene column, and the flow-through was collected as fraction 1. The beads were washed once more with 1 mL of dialysis buffer, and the flow-through was collected as fraction 2. The two fractions were analyzed by resolving on 4–20% gradient SDS-PAGE and stained with Roti Blue. Protein concentration was measured using NanoDrop and stored at −80 °C.

For His-tagged protein purification, the cells were resuspended in lysis buffer (50 mM Tris-HCl at pH 7.4, 150 mM NaCl, 5 mM MgSO$_4$, 4 mM DTT, 100 mM β-lactose, 100 μM phenylmethylsulfonyl fluoride) with ~5 mg of DNAseI (Merck, #10104159001) and ~5 mg of lysozyme (Thermo Fisher Scientific, #89833). Cells were lysed using an LM10 microfluidizer (Microfluidics, USA) at 18000 PSI, and cell debris was separated by centrifugation (4 °C, 30 min, 75,600×g, JA25.50 rotor Beckman Coulter). The supernatant was loaded on an affinity chromatography column (GE Healthcare, His-Trap FF crude, #17-5286-01) with a flow rate of 1 mL/min. A total amount of 10 column volumes 10% elution buffer (50 mM Tris-HCl pH 7.4, 150 mM NaCl, 5 mM MgSO$_4$, 100 mM β-lactose, 4 mM DTT, 1 M Imidazole) and 90% lysis buffer (50 mM Tris-HCl pH 7.4, 150 mM NaCl, 5 mM MgSO$_4$, 4 mM DTT, 100 mM β-lactose) with a flow rate of 2 mL/min was applied. The protein was then eluted using 5 column volumes of elution buffer (50 mM Tris-HCl pH 7.4, 150 mM NaCl, 5 mM MgSO$_4$, 100 mM β-lactose, 4 mM DTT, 1 M Imidazole). Afterwards, the protein was injected into a size exclusion chromatography system (GE Healthcare, HiLoad 16/600 Superdex 75 pg, #28-9893-33) using SEC buffer (20 mM HEPES pH 7.4, 150 mM NaCl, 5 mM MgSO$_4$, 100 mM β-lactose, 4 mM DTT) and a flow rate of 1 mL/min. Protein-containing fractions were pooled, concentrated (MWCO = 3 kDa) to 16.1 mg/mL, snap-frozen in liquid nitrogen, and stored at −80 °C. The protein concentration was measured using Nano-Drop 2000c Spectrophotometer (Thermo Fisher Scientific).

### Fluorescence polarization assays

The fluorescence polarization assay was adapted from our previously established protocol[62,67]. The non-labeled L5UR and their derivatives and FITC-labeled peptides were obtained from Pepmic Co., China. F-L5UR was synthesized by attaching fluorescein to the N-terminus amino group, leucine of L5UR peptide via aminohexanoic acid linker.

For the direct binding assay, the GST-B-RBD, or GST, was 2-fold diluted in an assay buffer composed of 50 mM Tris HCl pH 7.4, 50 mM NaCl, 5 mM DTT, and 0.005% (v/v) Tween 20 in a black low volume, round bottom 384-well plate (Corning, #4514). Then 10 nM F-L5UR peptide was added to each well and incubated for 20 min at ~22 °C on a horizontal shaker. The fluorescence polarization measurement was performed on the Clariostar (BMG Labtech) plate reader, using a fluorescence polarization module ($\lambda_{excitation}$ 482 ± 8 nm and $\lambda_{emission}$ 530 ± 20 nm). The milli fluorescence polarization, mP, was determined from the measured fluorescence intensities, calculated according to,

$$mP = 1000 \times \frac{I_h - I_v}{I_h + I_v}$$

where $I_v$ and $I_h$ are the fluorescence emission intensities detected with vertical and horizontal polarization, respectively. The mP was plotted

against the concentration of the GST-RBD and the $K_D$ value of the F-L5UR was calculated using a quadratic equation,

$$y = \frac{Af + (Ab - Af) * (Lt + K_D + x - \sqrt{(Lt + K_D + x)^2 - 4 * Lt * x}}{2Lt}$$

$Af$ is the polarization value of the free fluoresce nt probe, $Ab$ is the polarization value of the fluorescent probe/protein complex, $Lt$ is the total concentration of the fluorescent probe, $K_D$ is the equilibrium dissociation constant, $x$ is total concentration of protein and y is measured polarization value[36,67]. $K_D$ is measured in the same unit as $x$. For competitive fluorescence polarization experiments, the non-labeled peptides were threefold diluted in the assay buffer and then a complex of 5 nM F-L5UR peptide and 200 nM RBD was added to the dilution series to a final volume of 20 μL per well in 384-well plate. After 30 min incubation at ~22 °C, the fluorescence polarization was read. The logarithmic concentration of peptide was plotted against the mP-value and the data were fit with the log (inhibitor) vs response four parameters equation in GraphPad, and the IC$_{50}$ values were derived. Some IC$_{50}$ values were converted into $K_D$ values as described earlier[68].

### QRET assays

The QRET assays were modified from our previously described quenching luminescence assays[69–71]. Ac-K-L5URcore was conjugated with nonadentate europium chelate, {2,2′,2′′,2′′′-{[4′-(4′′′-iso-thiocyanatophenyl)-2,2′,6′,2′′-terpyridine-6,6′′-diyl]bis(methylene-nitrilo)}tetrakis(acetate)}europium(III) (QRET Technologies, Finland) via the epsilon amine of the N-terminal lysine that was added to the L5UR-core peptide sequence and purified with analytical reverse-phase HPLC. The current homogeneous QRET binding assay is based on the quenching of non-bound Eu-K-L5URcore with MT2 quencher (QRET Technologies), while bound labeled peptide is luminescent. In the assay, purified B-RBD or Gal1 were twofold diluted in an assay buffer containing 10 mM HEPES pH 7.4 and 10 mM NaCl added in 5 μL to a white low-volume, round bottom 384-well plate. Eu-K-L5UR core peptide (29 nM), mixed with MT2 according to the manufacturer's instructions in the assay buffer supplemented with 0.01% (v/v) Triton X-100, was added in 5 μL volume to wells, and incubated for 30 min at ~22 °C on a shaker. The luminescence was measured with Tecan Spark multimode microplate reader (Tecan, Austria) in time-resolved mode using $\lambda_{excitation}$ 340 ± 40 nm and $\lambda_{emission}$ 620 ± 10 nm with 800 μs delay and 400 μs window times.

### Circular dichroism spectra

Acetylated peptides were dissolved in buffer (1× PBS pH 7.5) to a final concentration of 25 μM. Measurements were performed using a Jasco Circular Dichroism spectrometer (J-1500) in a quartz cuvette (1 mm pathlength, Hellma) at 20 °C. Spectra were recorded in 5 continuous scans at a scanning speed of 100 nm min$^{-1}$ (1 mdeg sensitivity, 0.5 nm resolution, 1.0 nm bandwidth, 2 s integration time). From each measurement, values from a blank control containing only the buffer were subtracted to obtain the final ellipticity (mdeg), which was transformed into the mean residue ellipticity (MRE/deg cm$^2$ dmol$^{-1}$).

### In vitro pull-down assays with recombinant proteins

Biotinylated L5UR (bio-L5UR) peptide was synthesized as described above with a PEG5-linker to link the biotin to the L5UR peptide. GST, GST-B-Raf-RBD (155–227), and His-Gal1 were prepared as described above. Each protein in the assay was used at 2 μM concentration, and the peptide was at 4 μM in a reaction of 150 μL. First, peptide and Gal1 were pre-incubated for 30 min at 37 °C, then GST-B-RBD or GST alone was added, and the reaction continued for another hour. Control reaction mixes contained DMSO-vehicle instead of the peptide. At the end of the reaction time, 10 μL of each sample was withdrawn for SDS-PAGE analysis as inputs. For pull-downs,

5 μL of the beads were taken per sample. To prepare the beads, an appropriate volume of the slurry was pipetted into 15 mL falcon tubes and centrifuged at 830×g for 1 min to remove the ethanol-containing supernatant. The falcon tube was topped up to 15 mL with distilled water and centrifuged for 1 min to remove water. This washing step was repeated three times. Finally, the beads were resuspended in distilled water so that the final bead volume was 4× diluted i.e., 20 μL were pipetted to each tube. Pull-down was conducted by incubating samples on a rotating wheel at room temperature (20–25 °C) for 1 h. Then, the samples were centrifuged for 1 min at 830×g at 4 °C. The supernatant was discarded, and the beads were rinsed with 250 μL of washing buffer (50 mM Tris HCl pH 7.5, 150 mM NaCl, 4 mM β-mercaptoethanol, 0.05% (v/v) NP-40, 10% (v/v) Glycerol) for the total of 1 h at 4 °C, with four exchanges of the washing buffer. The bound material was eluted off the beads by adding 2× SDS-PAGE sample buffer and incubating for 5 min at 95 °C. The analysis was done by resolving the samples (8 μL of the input samples and 10 μL of the eluted material) on 4–20% gradient SDS-PAGE gels and analyzed by Western blotting. A list of all the antibodies used in the study and their sources are given in Supplementary Table 1.

## SNAP-tag mediated pull-downs

For the pull-down of interactors of L5UR-SNAP and control constructs, HEK293 EBNA cells were plated on 10 cm dishes. For each dish, 5 μg of pDest305-CMV-hGal1 and pEF-L5UR-SNAP, pEF-mutL5UR-SNAP, or pEF-SNAP were transfected. Transfection was done at ≥70% confluency using 2 μL jetPRIME per 1 μg DNA transfected, according to the manufacturer's instructions. After about 24 h, the growth medium was removed, cells were rinsed twice in cold PBS, and each dish was lysed in 1 mL lysis buffer, consisting of 20 mM HEPES pH 7.5, 4 mM β-mercaptoethanol, 0.05% Igepal, 1× Protease Inhibitor (Thermo Scientific, #A32955) and 1× PhosSTOP (Roche, #04 906 837 001). The cells were scraped and transferred to Eppendorf tubes, then incubated on ice for 30 min, with occasional mixing by inverting the tube. The lysate was cleared by centrifugation for 15 min at 4 °C and 16,363 rcf. Cleared lysate was transferred to a clean tube, 15 μL sample was withdrawn as "Input" for Western blot analysis, and 25 μL of SNAP-capture magnetic beads (New England Biolabs, #S9145S) suspension (diluted 1:1 in lysis buffer) was added. The samples were further incubated for 2 h at room temperature (20–25 °C) on a rotating wheel. Next, the supernatant was discarded, and the beads were rinsed 3× 10 min with 1 mL of washing buffer consisting of 50 mM Tris pH 7.5, 150 mM NaCl, 2 mM EDTA, 2 mM DTT, 0.5% NP-40, 1× Protease Inhibitor, and 1× PhoSTOP. The bound material was released off the beads by adding 25 μL of 2× SDS-PAGE sample buffer and incubating for 5 min at 95 °C. The samples were resolved on 4–20% SDS-PAGE gels in Tris-Gly buffer and analyzed by Western blotting. A list of all the antibodies used in the study and their sources are given in Supplementary Table 1.

## Electron microscopic analysis of Ras-nanoclustering

To quantify the nanoclustering of a component integral to the plasma membrane (PM), the apical PM sheets of baby hamster kidney (BHK) cells expressing a GFP-tagged H-Ras construct were fixed with 4% (w/v) PFA and 0.1% (w/v) glutaraldehyde. GFP anchored to the PM sheets was probed with 4.5 nm gold particles pre-coupled to anti-GFP antibody. Following embedment with methyl cellulose, the PM sheets were imaged using transmission electron microscopy (JEOL JEM-1400). Using the coordinates of every gold particle, Ripley's K-function calculated the extent of nanoclustering of gold particles within a selected 1 μm² PM area:

$$K(r) = \mathrm{A}n^{-2}\sum_{i \neq j} w_{ij}1(\|x_i - x_j\| \leq r)$$

$$L(r) - r = \sqrt{\frac{K(r)}{\pi}} - r$$

where $n$ gold particles populate in an intact area of $A$; $r$ is the length between 1 and 240 nm; $\|\cdot\|$ indicates Euclidean distance where $1(\cdot) = 1$ if $\|x_i - x_j\| \leq r$

and $1(\cdot) = 0$ if $\|x_i - x_j\| > r$; $K(r)$ specifies the univariate K-function. $w_{ij}^{-1}$ is a parameter used for an unbiased edge correction and characterizes the proportion of the circumference of a circle that has the center at $x_i$ and radius $\|x_i - x_j\|$. Monte Carlo simulations estimate the 99% confidence interval (99% C.I.), which is then used to linearly transform $K(r)$ into $L(r) - r$. On a nanoclustering curve of $L(r) - r$ vs. $r$, the peak $L(r) - r$ value is used as summary statistics for nanoclustering and is termed as $Lmax$. For each condition, at least 15 PM sheets were collected for analysis. To analyze statistical significance between conditions, bootstrap tests compare our point patterns against 1000 bootstrap samples.

## Immunoblotting

Routinely, 4–20% Mini-PROTEAN TGX Precast Protein Gels, 10-well, 50 μL, or 30 μL (BioRad, #4561094 or #4651093) were used, unless stated otherwise. For protein size reference, Precision Plus Protein All Blue Prestained Protein Standards (BioRad, #1610373) or Page Ruler Prestained (Thermo Fisher Scientific, #26616) were used. For ERK activity studies, Hs 578 T, T24, MIA PaCa-2 and HEK cells were grown in a 6-well plate for 24 h. After 16 h serum starvation, the cells were treated for 2 h with the L5UR derived TAT-peptides or DMSO control, before they were stimulated with 200 ng/ mL EGF for 10 min. The cell lysates were then prepared using a buffer composed of 150 mM NaCl, 50 mM Tris-HCl pH 7.4, 0.1% (w/v) SDS, 1% (v/v) Triton X-100, 1% (v/v) NP40, 1% (w/v) Na-deoxycholate, 5 mM EDTA pH 8 and 10 mM NaF completed with 1× protease inhibitor cocktail (Pierce, #A32955) and 1× phosphatase inhibitor cocktail (Roche PhosSTOP, #490684001). The total protein concentration was determined using Bradford assay (Protein Assay Reagent, BioRad, #5000006) and 25 μg cell lysate was loaded on a 10% homemade SDS-PAGE gel.

For immunoblotting, gels were transferred onto 0.2 μm pore-size nitrocellulose membrane by using Trans-Blot Turbo RTA Midi 0.2 μm Nitrocellulose Transfer Kit, for 40 blots (BioRad, #1704271). The membranes were blocked with TBS or PBS with 0.2% (v/v) Tween20 and 2% BSA. Primary antibodies were incubated at 4 °C for 16 h or for 1–3 h at room temperature (20–25 °C). All secondary antibodies were diluted at 1:10,000 in a blocking buffer and were incubated for 1 h at room temperature (20–25 °C). A detailed list of all the antibodies used in the study and their sources are given in Supplementary Table 1.

## Fluorescence lifetime imaging microscopy (FLIM)-FRET analysis

FLIM-FRET experiments were conducted as described previously[29,30,72]. About 120,000 HEK cells were seeded per well in a 6-well plate (Greiner, #657160) with a cover slip (Carl Roth, #LH22.1) and grown for 18–24 h. For H-RasG12V nanoclustering-FRET, the cells were transfected with a total of 1 μg of mGFP/mCherry-tagged H-RasG12V at a donor (D):acceptor (A)-plasmid ratio of 1:3. In addition, 0.75 μg of other plasmids encoding L5UR, rat (rt) Gal1 or N-rtGal1 (dimerization-deficient mutant) were co-transfected. For Gal1/C-RBD FRET-interaction, the cells were transfected with 2 μg mGFP-rtGal1 and mRFP-C-RBD (D:A, 1:3) or mGFP-rtGal1 and mRFP-C-RBD-D117A pair (D:A, 1:3). In addition, cells were co-transfected with 1.5 μg pClontech-C-L5UR, the pcDNA-Hygro-Anginex or compound OTX008 (Cayman Chemicals, #23130). All transfections were done using jetPRIME (Polyplus, #114-75) transfection reagent according to the manufacturer's instructions. After 4 h of transfection the medium was changed. The next day, the cells were fixed with 4% w/v PFA. The cells were mounted with Mowiol 4–88 (Sigma-Aldrich, #81381). An inverted microscope (Zeiss AXIO Observer D1) with a fluorescence lifetime imaging attachment (Lambert Instruments) was used to measure fluorescence lifetimes of mGFP. Fluorescein (0.01 mM, pH 9) was used as a fluorescence lifetime reference ($\tau$ = 4.1 ns). Averaged fluorescence lifetimes were used to calculate the apparent FRET efficiency as described[30,72].

## Split-luciferase KinCon B-Raf biosensor measurements

HEK 293-EBNA cells were seeded in a 12-well plate (Greiner Bio-One, #665180) in 1 ml complete DMEM and grown for 24 h. The next day 0.5 μg

of KinCon sensor plasmid pcDNA-RlucF1-BRAF-RlucF2 was transfected along with 0.1–0.5 µg of modulator plasmid encoding either SNAP-H-RasG12V or Gal1) using jetPRIME as per manufacturer protocol; pcDNA3.1(−) Thermo Fisher Scientific, #V79520) was used to buffer the total amount of plasmid load per well to 1 µg. After 48 h of expression cells were collected and washed in PBS. Cells from one well of the 12-well plate were resuspended in 200 µL of PBS and 2× 90 µL were pipetted into a white 96-well plate (Nunc, Thermo Fisher Scientific, #236108). Then coelenterazine h was added to a final concentration of 5 µM and the luminescence signal at 480 ± 10 nm was collected for 10 s. The basically background-free signal was normalized against the signal without modulator plasmid.

### BRET assays

We employed the BRET2 system where RLuc8 and GFP2 luminophores were predominantly used as the donor and acceptor, respectively, with coelenterazine 400a as the substrate. A CLARIOstar plate reader from BMG Labtech was used for BRET and fluorescence intensity measurement. The BRET protocol was adapted as described by us[73].

In brief, 150,000–200,000 HEK293-EBNA cells were seeded per well of a 12-well plate and grown for 24 h in 1 mL of complete DMEM. The next day, the cells were transfected with ~1 µg of plasmid DNA per well using a 3 µL jetPRIME transfection reagent. For the donor saturation titration, 25 ng of the donor plasmid was transfected with an acceptor plasmid concentration ranging from 25 to 1000 ng. The pcDNA3.1(−) plasmid (was used to top up the amount of DNA per well. 48 h after transfection, cells were collected in PBS and plated in a white 96-well plate.

First, the fluorescence intensity of GFP2 was measured ($\lambda_{excitation}$ 405 ± 10 nm and $\lambda_{emission}$ 515 ± 10 nm), which is directly proportional to the acceptor expression (RFU). Then 10 µM of coelenterazine 400a (GoldBio, #C-320) was added to the cells, and BRET readings were recorded simultaneously at $\lambda_{emission}$ 410 ± 40 nm (RLU) and 515 ± 15 nm (BRET signal). Emission intensity measured at 410 nm is directly proportional to the donor expression. The raw BRET ratio was calculated as the ratio of BRET signal/RLU. The background BRET ratio was obtained from cells expressing only the donor. The background BRET ratio was subtracted from the raw BRET ratio to obtain the BRET ratio, plotted here as 'BRET'. The relative expression was calculated as the ratio of RFU/RLU. The relative expression, acceptor/ donor, plotted in the x-axis in corresponding figures, was obtained by normalizing RFU/RLU values to those from cells transfected with 1:1 donor and acceptor plasmid ratio[54].

Alternatively, the fluorescence intensity of mNeonGreen was measured at $\lambda_{excitation}$ 485 ± 10 nm and $\lambda_{emission}$ 535 ± 10 nm. Then 2.9 µM of coelenterazine 400a was added to the cells and the BRET readings for mNeonGreen and NanoLuc were recorded simultaneously at $\lambda_{emission}$ 460 ± 25 nm (RLU) and 535 ± 25 nm (BRET signal).

The BRET ratio and acceptor/donor values from various biological repeats were plotted together and the data were fitted with a hyperbolic equation in Prism (GraphPad). The one phase association equation of Prism 9 (GraphPad) was used to predict the top asymptote Ymax-value, which was taken as the BRETtop. The BRETtop value represents the top asymptote of the BRET ratio reached within the defined acceptor/donor range.

For the dose-response BRET assays, the donor and acceptor plasmid concentration were kept constant, as indicated in the corresponding figure legends. HEK293-EBNA cells were grown in 12-well plate for 24 h in complete DMEM. The next day, donor and acceptor plasmids were transfected along with modulator plasmid ranging from 125 to 850 ng. After 48 h of expression the cells were collected in PBS and BRET measurements were carried out.

For treatment with peptides, HEK cells were batch-transfected. After 24 h of transfection, cells were re-plated in a white 96-well plate in phenol red-free DMEM. After another 48 h, peptides were added to cells at concentrations ranging from 0.1 µM to 100 µM. After 2 h incubation at 37 °C, the plate was brought to room temperature (20–25 °C) before taking BRET measurements as indicated above. The concentration of the

transfected L5UR-modulator plasmid or applied peptide was plotted against the BRET value and the data were fitted with a straight-line equation using Prism.

### Cell viability assay and drug sensitivity score (DSS) analysis

The cells were seeded in low attachment, suspension cell culture 96-well plates (Greiner, #655185). About 2000 T24, MIA PaCa-2, and HEK cells and 5000 Hs 578 T cells were seeded per well in a 50 µL complete growth medium. 24 h later, the cells were treated with 50 µL 2× peptide diluted in the growth medium or 0.2% (v/v) of the positive control, benzethonium chloride stock at 100 mM in $H_2O$ (Sigma-Aldrich, #B8879). Forty-eight hours after the peptide treatment 10% (v/v) of alamarBlue reagent (Thermo Fisher Scientific, #DAL1100) was added to each well and incubated for 4 h at 37 °C. Using a CLARIOstar plate reader the fluorescence signal ($\lambda_{excitation}$ 560 ± 5 nm and $\lambda_{emission}$ 590 ± 5 nm) was recorded. The florescence signal was normalized against the negative control, here DMSO in buffer, representing 100% viability. Additionally, the data was analyzed using Breeze 2.0 to determine a drug sensitivity score (DSS), a normalized area under the curve (AUC). Here we plot only one of the output values from the Breeze pipeline[74], the DSS3 value, which was calculated as

$$DSS_3 = DSS_2 \frac{x_2 - x_1}{C_{max} - C_{min}}$$

where $DSS_2$ is given by the equation $DSS_2 = \frac{DSS_1}{\log a}$

And $DSS_1$ is given by the equation $DSS_1 = \frac{AUC - t(x_2 - x_1)}{(100 - t)(C_{max} - C_{min})}$

After dose–response inhibition data fitting with a logistic function, the area under the curve (AUC) was determined. The activity threshold (t) was set to ≥10%. The maximum ($C_{max}$) and minimum ($C_{min}$) concentrations used for screening of the inhibitors, with $C_{max} = x_2$ and $x_1$ concentration with minimal activity t. The parameter a is the value of the top asymptote, which can be different from 100% inhibition as obtained from the benzethonium chloride positive control value.

### Statistics and reproducibility

Data were analyzed using Graph Pad prism 9.0 software. The number of independent biological repeats (n) for each dataset is provided in the figure legends. If not stated otherwise means and standard errors (SEM) are plotted. The statistical significance of differences between Lmax-values determined in the nanoclustering analysis by electron microscopy was determined using bootstrap tests. All BRETtop data were compared using the extra sum-of-squares F test. All other statistical analyses were performed using one-way ANOVA. A p-value of <0.05 was considered statistically significant, and the statistical significance levels were annotated.

### Reporting summary

Further information on research design is available in the Nature Portfolio Reporting Summary linked to this article.

## Data availability

All data supporting the findings of this study are available within the manuscript and its Supplementary Information. Uncropped and unedited blot images with references to respective figures are provided in Supplementary Figs. 4–19. All source data for graphs in this manuscript are provided in Supplementary Data 1. All unique/stable reagents generated in this study are available from the corresponding author with a completed materials transfer agreement. This study did not report standardized datatypes.

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

## Acknowledgements

The study was supported by the grant INTER/NWO/19/14061736-HRAS-PPi of the Luxembourg National Research Fund (FNR) and OCENW.KLEIN.228 of the Dutch Research Council (NWO) to DA and TG, respectively, as well as FNR-grant INTER/Mobility/2021/BM/15591725/panRAFi-PB to GM. Prof. Marc Therrien (IRIC, Université de Montréal, Canada) is gratefully acknowledged for hosting GM in his laboratory. Dr. Hugo Lavoie and Dr. Ting Jin (IRIC, Université de Montréal, Canada) are thanked for their support and advice to GM during his sabbatical. Geneviève Arseneault (IRIC, Université de Montréal, Canada) is thanked for the technical support. pET28a-L5UR was a gift from Dr. Latifa Elantak (CNRS Marseille, France). pcDNA-Hygro-Anginex was a gift from Prof. Victor L. Thijssen (VU Amsterdam, Netherlands). The KinCon plasmid was obtained from Dr. Eduard Stefan (University of Innsbruck, Austria).

## Author contributions

D.K.A. and T.N.G. conceived the study. C.L.S. collected and evaluated BRET, FP, signaling, and cell viability data and purified proteins. G.M. collected and evaluated BRET, FP, and KinCon data. K.P. purified proteins and performed and evaluated pull-down experiments and signaling data. A.Y.V. synthesized all of the peptides in Supplementary Table 2 and performed CD spectra. MK collected and evaluated FLIM-FRET data. Y.Z. and N.A. collected and evaluated EM-nanoclustering data. H.H. collected and evaluated QRET data. A.G. performed bioinformatics analysis of survival and cancer type frequency. T.N.G. and D.K.A. jointly supervised the study. G.M., C.L.S., K.P., A.Y.V., T.N.G., and D.K.A. wrote the paper.

## Competing interests

The authors declare no competing interests.
