## [Peer Review File · Communications Biology]

Reviewers' comments:

Reviewer #1 (Remarks to the Author):

This paper reports the discovery that a 23 amino acid peptide, corresponding to the L5UR core derived from the Galectin-1 (Gal1) binding Pre-B Cell Receptor, decreases Gal1-Raf interaction and Ras nanoclustering, which is a requirement for signaling through the MAPK pathway. The study uses the fluorescence-based assay BRET with RLuc8 donor and GFP2 acceptor tags to detect protein/protein interactions in cells. Through this method, the authors establish that interaction between Gal1 and RBD and Ras-Ras interactions (nanoclustering) are both significant and that the effect is greatest with B-RBD. They then use FRET between mGFPGal-1 and mRFP-C-RBD to show that the L5UR peptide interferes with the Gal1-C-RBD interaction and with Ras nanoclustering in cells. These effects were shown to depend on the dimeric form of Gal1. In order to determine that L5UR peptide engages directly at the binding interface of Gal1 with RBD, the authors used pull-down assays with purified His-tagged-Gal1 and GST-RBD. They used biotinylated-L5UR to pull down Gal1 and GST-RBD, and then used a fluorescence polarization assay to determine that the L5UR interacts with RBD with micromolar affinity, as does Gal1 (determined previously by NMR). They also did the experiments with a series of positively charged residues on L5UR mutated to negatively charged residues, and showed that the mutant peptide has no effect, concluding that the L5UR-RBD interaction is electrostatic in nature. Next, the authors use a cell penetrating version of L5UR to show that the disruption by the peptide of the Gal1-RBD interaction also occurs in cells as does the decrease in Ras nanoclustering. Finally, making use of cancer cell lines they showed that L5UR disrupts MAPK signaling and inhibits H-Ras-mutant cancer cell proliferation.

The paper provides a nice series of experiments that validate the Gal1-RBD interface as a drug target and L5UR as a starting precursor to disruption of signaling through the MAPK pathway. However, interpretation of the data in terms of specific structural models should be done with caution, both in terms of support for the stacked dimer model and in terms of the mode of interaction of the peptide with RBD. For example, the conclusion that the interaction between the peptide and RBD must be electrostatic in nature does not necessarily follow from the fact that the charge reversal mutant peptide does not interact with RBD. This implies the assumption that the peptide remains helical which very likely is not the case, which would be expected to impact interaction due to greater disorder in the peptide. It looks like the model in figure S2 was created in PyMol without any testing and it could be very misleading. It should be either tested, as the authors experimentally tested their putative binding site for the Gal1-RBD interaction (2016) or removed from the paper. Most of the positively charged residues mutated to negatively charged residues in the L5UR for these experiments were previously shown to be at the binding interface with Gal1 (obtained based on NMR data). The cell results with L5UR could be due to its interaction with Gal1 rather than with RBD, and the lack of binding to Gal1 due to the charge reversal mutations would be expected to give the cell-based results presented here.

Reviewer #2 (Remarks to the Author):

Comments:

Hudalla and colleagues (PMID: 34777608) pointed out that the promise of the natural immunoregulator, Galectin-1 (Gal1), in therapeutic has been challenged by its unstable homodimeric conformation. In earlier experiments the homodimer was stabilized via covalent poly(ethylene glycol) diacrylate (PEGDA) cross-linking, which then demonstrated higher activity relative to the non-covalent homodimer. The fact that it is unstable agrees with the authors earlier work, where they used higher concentrations of Gal1. At higher concentration it can dimerize in vitro. Considering the low affinity, the question then is to what extent Raf's activation relies on Gal1, and consequently how inhibition of Gal1-RBD would impact Raf activation.

Here the aims are to further confirm the authors' earlier results about Gal1 scaffolding and to demonstrate its usefulness as a drug target. The first aim is accomplished. As to whether it can be a practical target, is still unclear.

Reviewer #3 (Remarks to the Author):

Manoharan and colleagues use BRET to measure the dimerization of galectin 1, interaction with RAF RBD and the formation of HRAS nanoclusters. The authors use a previously identified peptide (L5UR), which is a region of the pre-B cell receptor and binds to galectin 1 (Gal1) with an affinity of 310 μ M, to interrogate the role Gal1 plays in HRAS nanocluster formation. Using biochemical assays, the authors demonstrate that L5UR binds with an increased affinity to RBD and can disrupt Gal1/RBD interaction in cells. Additional FRET experiments suggest that L5UR decreases HRAS nanoclustering. Delivery of the peptide as a TAT fusion decreases MAPK signaling in cell lines expressing HRAS and Gal1 (Hs 578T and T24) but not KRAS/Gal1 (Mia PaCa2) or cells lacking Gal1 (HEK). Decrease in proliferation was seen with all cells with the L5UR but not a mutated version of the peptide suggesting that the proliferative defect was not driven by inhibition of the MAPK pathway. Overall, the effect of inhibition of Gal1/RBD and HRAS nanoclustering by L5UR is modest (doesn't reach 50%) at concentration of 50 - 100 μ M and does not correlate with the inhibition of the MAPK pathway (less than 50% inhibition at 20 μ M). This reviewer remains unconvinced that the mode of inhibition of MAPK inhibition is based on the disruption of Gal1/RBD interaction and HRAS nanoclustering. Unfortunately, the manuscript is missing several important controls that are required prior to publication.

1. Figure 4b shows 100 μ M L5UR inhibits Gal1/BRAF RBD interaction by about 25%; Figure 4c shows 50 μ M inhibits HRAS G12V interactions about 40% whereas the control mutL5UR peptide does not impact the activity. However, pERK is inhibited by approximately 40% at lower concentrations (Fig 4d and e), while the control mutL5UR peptide actually increases the pERK signal in T24, Mia PaCa-2 and HEK cells. The anti-proliferative activity of L5UR on HEK and Mia PaCa-2 suggests that the anti-proliferative activity is not due to inhibition of the MAPK pathway or via disruption of HRAS nanoclustering. While the authors admit that the TAT-L5URcore also interferes with other signaling

pathways they do not evaluate the activity of the peptide on other signaling pathways such the PI3K pathway. Further analysis of the peptide on this pathway is required and a more careful explanation of the discrepancy between the activity of the peptide in the various cell-based assays is required to convince this reviewer of the proposed mechanism of action.

2. The authors measure the affinity of L5UR to BRAF GST-RBD using fluorescence polarization. The affinity is an increase is 40-fold over the affinity of this peptide to Gal1 – this is a significant enhancement, do the authors have an explanation for this? In the pull-down experiments the peptide seems to be able to pull-down both GST-RBD and Gal1 to a similar level – this seems surprising given the significantly weaker affinity for Gal1. In the pull-downs containing both Gal1 and GST-RBD both proteins are pulled down to equal levels, again this is surprising as one would expect that the higher binding affinity to GST-RBD would result in less Gal1 being pulled down as more of the peptide would be bound to GST-RBD. Could the authors explain these results? In addition, the authors should repeat the pull-down experiments using the mutL5UR. A comparison of the affinity of the L5UR for Gal1 in their FP or the QRET assay should be included to provide further confidence in the data.

3. In the competition FP experiments (Fig 2f) the mP of the F5UR/CRAF-RBD (not BRAF RBD??) is around 220 and with disruption decreases to ~100 mP. Yet in Fig 2d the unbound peptide has a mP of 220. In the competition assay in S2e, the complex starts with a value of 150 mP – why the discrepancy? The equation used to fit the PF data include the free anisotropy and bound anisotropy values – presumably this should be free and bound polarization values?

4. The mutL5UR removes 4 hydrophobic amino acids, and 2 basic amino acids and replace with 5 acid residues. The L5UR peptide forms a short helix, what impact does incorporation of these mutations have on the structure of the peptide?

5. The effect of the L5UR peptide on the HRAS BRET signal is modest and while the mutant peptide shows no activity is difficult to evaluate the potency. How much would the BRET signal decrease if untagged HRAS-G12V were expressed to compete for the BRET signal. Jung et al (PMID: 30787043) determined that TRPMK1 attenuates HRAS nanoclustering by disrupting distribution and levels of cholesterol. Inhibitors of TRMK1, such as ML-SI1 show a decrease in HRAS nanoclustering. As Dr. Zhou is an author on both manuscripts it would be interesting to compare the activity of ML-SI1 in the HRAS BRET assay.

6. The authors show (Fig 1d) and state that Gal1 binds BRAF RBD, yet that use CRAF RBD to demonstrate the activity of L5UR at decreasing the interaction with Gal1 (Fig 2a), but than revert back to GST-BRAF RBD in their fluorescence polarization and pull-down assays. \Figure 2f uses CRAF-RBD (although the methods state BRAF RBD was used) in the FP competition assay. In BRET measurements (Figure 3b) BRAF-RBD is used again, but the modeling of the binding to RBD is done with CRAF. Given the larger signal observed in the Gal1/RBD BRET assay with BRAF RBD and the apparent specificity for BRAF RBD why do the authors switch the source of the RBD in the manuscript? Frankly this just causes confusion and a lack of clarity in the hypothesis being evaluated.

7. In BRET experiments measuring HRAS nanoclustering (Fig 2b) the authors show a modest increase in BRET signal with Gal1, the authors should include experiments showing the effect of N-Gal1, to provide further support for their model.

Point-by-point response to revision on manuscript COMMSBIO-23-3781-T: "Identification of an H-Ras nanocluster 1 disrupting peptide"

Reviewer 1:

This paper reports the discovery that a 23 amino acid peptide, corresponding to the L5UR core derived from the Galectin-1 (Gal1) binding Pre-B Cell Receptor, decreases Gal1-Raf interaction and Ras nanoclustering, which is a requirement for signaling through the MAPK pathway. The study uses the fluorescence-based assay BRET with RLuc8 donor and GFP2 acceptor tags to detect protein/protein interactions in cells. Through this method, the authors establish that interaction between Gal1 and RBD and Ras-Ras interactions (nanoclustering) are both significant and that the effect is greatest with B-RBD. They then use FRET between mGFPGal-1 and mRFP-C-RBD to show that the L5UR peptide interferes with the Gal1-C-RBD interaction and with Ras nanoclustering in cells. These effects were shown to depend on the dimeric form of Gal1. In order to determine that L5UR peptide engages directly at the binding interface of Gal1 with RBD, the authors used pull-down assays with purified His-tagged-Gal1 and GST-RBD. They used biotinylated-L5UR to pull down Gal1 and GST-RBD, and then used a fluoresce polarization assay to determine that the L5UR interacts with RBD with micromolar affinity, as does Gal1 (determined previously by NMR). They also did the experiments with a series of positively charged residues on L5UR mutated to negatively charged residues, and showed that the mutant peptide has no effect, concluding that the L5UR-RBD interaction is electrostatic in nature. Next, the authors use a cell penetrating version of L5UR to show that the disruption by the peptide of the Gal1-RBD interaction also occurs in cells as does the decrease in Ras nanoclustering. Finally, making use of cancer cell lines they showed that L5UR disrupts MAPK signaling and inhibits H-Ras-mutant cancer cell proliferation.

Question 1-1.

The paper provides a nice series of experiments that validate the Gal1-RBD interface as a drug target and L5UR as a starting precursor to disruption of signaling through the MAPK pathway. However, interpretation of the data in terms of specific structural models should be done with caution, both in terms of support for the stacked dimer model and in terms of the mode of interaction of the peptide with RBD. For example, the conclusion that the interaction between the peptide and RBD must be electrostatic in nature does not necessarily follow from the fact that the charge reversal mutant peptide does not interact with RBD. This implies the assumption that the peptide remains helical which very likely is not the case, which would be expected to impact interaction due to greater disorder in the peptide.

Response 1-1:

We thank the reviewer for the appreciative feedback on our manuscript and agree with a more cautious presentation of structural models.

Structural predictions by AlphaFold2 suggested a mostly unstructured L5URcore peptide (not shown). This is supported by experimental circular dichroism data now provided as **new Figure S2g**, which suggest that the peptide is mostly random coil with ~25% antiparallel β -sheet as the major secondary structure (L. 198). Even though, many peptides only become structured, once bound to their target surface, we no longer argue based on any helical secondary structure.

In the revised manuscript, we have therefore been more careful when explaining the charge reversal mutations in the peptide (L. 191):

The L5UR has a high proportion of six positively charged arginine residues in its core region, which may indicate that binding of the peptide to the RBD of Raf is influenced by electrostatic interactions. We therefore introduced several negatively charged, acidic residues to mostly replace basic and hydrophobic residues in the core-region of the L5UR peptide to generate a non-binding mutant (mutL5UR) (Figure 2f).

In addition, the claims in support of the stacked dimer model and other absolute structural models have now been removed or moderated in the revised main text at multiple positions. See also in the following for specific changes.

Question 1-2.

It looks like the model in figure S2 was created in PyMol without any testing and it could be very misleading. It should be either tested, as the authors experimentally tested their putative binding site for the Gal1-RBD interaction (2016) or removed from the paper.

Response 1-2:

We agree with the reviewer's concern and have removed Figure S2d, which was merely a computational model generated from an overlay of structures and structural models, as described. Unfortunately, our crystallography efforts for the L5UR-structure bound to the RBD of Raf did not succeed.

Question 1-3.

Most of the positively charged residues mutated to negatively charged residues in the L5UR for these experiments were previously shown to be at the binding interface with Gal1 (obtained based on NMR data). The cell results with L5UR could be due to its interaction with Gal1 rather than with RBD, and the lack of binding to Gal1 due to the charge reversal mutations would be expected to give the cell-based results presented here.

Response 1-3:

We agree that multiple targets of the L5UR could be responsible for its cellular effects. This is now reinforced by our new pull-down data showing that L5UR-SNAP does under those conditions not interact with Gal1, but several RBD- and RA-domain signaling proteins (**new Fig. 3e**). The interpretation of signaling and proliferation data has been adapted accordingly e.g., in the discussion (L. 291):

However, the broad impact on cell proliferation (Figure 6) and its engagement of several Ras interactors (Figure 3e), and its mixed effect on signalling (Figure 5), suggest that L5URcore is still an immature tool reagent.

The reviewer is right that all mutations are in the core region of L5UR and thus at the NMR-data predicted interface with Gal1. However, all new interactors discriminate between the L5UR-SNAP and mutL5UR-SNAP.

Moreover, the NMR-based Gal1-interaction was reported to be very weak ($K_D=310 \mu\text{M}$), as we state in our introduction and as supported by our new QRET-affinity data (**new Fig. S2e**).

Moreover, H-Ras nanoclustering is also reduced in HEK cells that are comparatively devoid of Gal1 expression (**Fig. S3d**), supporting that Gal1 is not critical for the L5UR effects.

Finally, we demonstrate the specific effect of L5UR as compared to another Gal1-binding peptide, Anginex. While Anginex can compete with L5UR for binding to Gal1 (**new Fig. S2b**), it does not disrupt binding of Gal1 and the C-Raf-RBD (**Fig. 2a**).

Reviewer 2:

Hudalla and colleagues (PMID: 34777608) pointed out that the promise of the natural immunoregulator, Galectin-1 (Gal1), in therapeutic has been challenged by its unstable homodimeric conformation. In earlier experiments the homodimer was stabilized via covalent poly(ethylene glycol) diacrylate (PEGDA) cross-linking, which then demonstrated higher activity relative to the non-covalent homodimer. The fact that it is unstable agrees with the authors earlier work, where they used higher concentrations of Gal1. At higher concentration it can dimerize in vitro.

Question 2-1.

Considering the low affinity, the question then is to what extent Raf's activation relies on Gal1, and consequently how inhibition of Gal1-RBD would impact Raf activation.

Here the aims are to further confirm the authors' earlier results about Gal1 scaffolding and to demonstrate its usefulness as a drug target. The first aim is accomplished. As to whether it can be a practical target, is still unclear.

Response 2-1:

We thank the reviewer for their feedback and recognition of our accomplishment. We agree and state in the first discussion paragraph that L5UR is still an immature tool reagent that inhibits the Raf-RBD/ Gal1 interaction.

Our BRET-data support that Gal1 can dimerize under our experimental conditions in HEK cells (**Fig. 1c**). We previously furthermore demonstrated a high affinity of 106 ± 40 nM for Gal1/ C-RBD (Blazevits O et al. PMID: 27087647), suggesting it may increase Raf-dimer stability by linking together two RBDs, at high Gal1 concentrations that permit its dimerization.

Raf-dimerization requires relieve of its autoinhibition, which opens its conformation as can be measured by KinCon biosensors. We now show that indeed Gal1 is able to open the conformation of B-Raf, consistent with its RBD-engagement (**new Fig. 1e**). We speculate at the end of that results paragraph (L. 145):

This could facilitate dimeric Ras-Raf engagement, which however requires a number of other modifications and conformational rearrangements.

We do not claim that Raf activation cannot progress without Gal1, but that Gal1 can be a modulator.

Reviewer 3:

Manoharan and colleagues use BRET to measure the dimerization of galectin 1, interaction with RAF RBD and the formation of HRAS nanoclusters. The authors use a previously identified peptide (L5UR), which is a region of the pre-B cell receptor and binds to galectin 1 (Gal1) with an affinity of 310 uM, to interrogate the role Gal1 plays in HRAS nanocluster formation. Using biochemical assays, the authors demonstrate that L5UR binds with an increased affinity to RBD and can disrupt Gal1/RBD interaction in cells. Additional FRET experiments suggest that L5UR decreases HRAS nanoclustering. Delivery of the peptide as a TAT fusion decreases MAPK

signaling in cell lines expressing HRAS and Gal1 (Hs 578T and T24) but not KRAS/Gal1 (Mia PaCa2) or cells lacking Gal1 (HEK). Decrease in proliferation was seen with all cells with the L5UR but not a mutated version of the peptide suggesting that the proliferative defect was not driven by inhibition of the MAPK pathway. Overall, the effect of inhibition of Gal1/RBD and HRAS nanoclustering by L5UR is modest (doesn't reach 50%) at concentration of 50 - 100uM and does not correlate with the inhibition of the MAPK pathway (less than 50% inhibition at 20uM). This reviewer remains unconvinced that the mode of inhibition of MAPK inhibition is based on the disruption of Gal1/RBD interaction and HRAS nanoclustering. Unfortunately, the manuscript is missing several important controls that are required prior to publication.

Question 3-1.

Figure 4b shows 100uM L5UR inhibits Gal1/BRAF RBD interaction by about 25%; Figure 4c shows 50uM inhibits HRAS G12V interactions about 40% whereas the control mutL5UR peptide does not impact the activity. However, pERK is inhibited by approximately 40% at lower concentrations (Fig 4d and e), while the control mutL5UR peptide actually increases the pERK signal in T24, Mia PaCa-2 and HEK cells.

The anti-proliferative activity of L5UR on HEK and Mia PaCa-2 suggests that the anti-proliferative activity is not due to inhibition of the MAPK pathway or via disruption of HRAS nanoclustering. While the authors admit that the TAT-L5URcore also interferes with other signaling pathways they do not evaluate the activity of the peptide on other signaling pathways such the PI3K pathway. Further analysis of the peptide on this pathway is required and a more careful explanation of the discrepancy between the activity of the peptide in the various cell-based assays is required to convince this reviewer of the proposed mechanism of action.

Response 3-1:

We thank this reviewer for their careful assessment and constructive feedback.

We agree that the results of the on-target BRET assays (**Fig. 4**) do not fully align with the phenotypic effects (signaling and proliferation, **Figs 5 and 6**) in cells. This discordance is common for such early stage, low potency tool compounds. This is now reflected in the revised first paragraph of our discussion, which incorporates new data that are explained more below (L. 291):

However, the broad impact on cell proliferation (Figure 6) and its engagement of several Ras interactors (Figure 3e), and its mixed effect on signalling (Figure 5), suggest that L5URcore is still an immature tool reagent.

Differences in the maximal effect also originate in the distinct dynamic ranges of the assays and differences in target protein expression. In the BRET-assay shown in **Fig. 4b** in particular, the target B-RBD has to be expressed in excess for an optimal dynamic range of the BRET-signal, hence the L5UR-effect is expected to be less. We now report EC50-values for the on-target BRET assays in **Fig. 4b,c** of $16 \pm 1 \mu\text{M}$ for Gal1/B-RBD and $19 \pm 1 \mu\text{M}$ for H-RasG12V nanoclustering, which agree with each other. These values are in line with effect concentrations observed in signaling and proliferation assays in Figures 5 and 6.

Regarding the effect of L5UR on multiple pathways, we are now providing new data suggesting a broader activity of L5UR. We show that it engages with several RBD- and RA-containing signaling proteins (**new Fig. 3e**). Thus, the phenotypic effects on signaling and proliferation are likely to be complex, and we now argue more carefully to not link these effects exclusively to the H-RasG12V nanocluster disruption (L. 291). Furthermore, we studied the effect of L5UR on pAKT downstream of active Ras in all four tested cell lines (**new Fig. 5e-h**). In brief, these data support that in cancer cell lines TAT-L5URcore suppresses relative pAKT-levels.

Question 3-2.

The authors measure the affinity of L5UR to BRAF GST-RBD using fluorescence polarization. The affinity is an increase is 40-fold over the affinity of this peptide to Gal1 – this is a significant enhancement, do the authors have an explanation for this?

In the pull-down experiments the peptide seems to be able to pull-down both GST-RBD and Gal1 to a similar level – this seems surprising given the significantly weaker affinity for Gal1. In the pull-downs containing both Gal1 and GST-RBD both proteins are pulled down to equal levels, again this is surprising as one would expect that the higher binding affinity to GST-RBD would result in less Gal1 being pulled down as more of the peptide would be bound to GST-RBD. Could the authors explain these results?

In addition, the authors should repeat the pull-down experiments using the mutL5UR.

A comparison of the affinity of the L5UR for Gal1 in their FP or the QRET assay should be included to provide further confidence in the data.

Response 3-2:

We thank the reviewer for pointing this out. We cannot explain why the affinity of the L5UR is higher to the Raf-RBD than to Gal1 and consider this finding serendipitous. The affinity to Gal1 was reported by Elantak et al. to be very low ($K_D=310 \mu\text{M}$), and a physiological relevance is not obvious based on this.

We understand the reviewer's consideration regarding the pull-down data (**Fig. 2c**) but would hesitate to quantitatively compare them to the fluorescence polarization binding experiments (**Fig. 2d, new Fig. S2d,e**, see below). The pull-down experiments are rather indirect, with capture and display of the peptide on the beads, which will affect its activity and how it can engage with each of the targets. As compared to other experiments, the biotin-tag of the L5UR is in this case at the N-terminus, which may also have an impact. Importantly, it is done with purified proteins, hence the peptide will not have to select between multiple, also non-binding targets.

In our new dataset showing pull-down experiments with several signaling proteins of the Ras pathway, we also include pull down data using the mutL5UR (**new Fig. 3e**). These data confirm a general loss of binding to any of these proteins when using the mutL5UR. This tentatively suggests that all these interactions involve the same, mutated stretch of residues in the L5URcore. These pull-down experiments do not recover Gal1, despite its overexpression, in agreement with its much lower affinity.

We furthermore performed additional binding experiments to compare affinities. While we find in the QRET an activity of $IC_{50}=18\pm 1 \mu\text{M}$ for L5URcore binding to B-RBD (**new Fig. S2d**), we do not find saturation at the highest technically possible concentration of $135 \mu\text{M}$ for L5URcore binding to Gal1 (**new Fig. S2e**), which agrees with the reported low affinity of $K_D=310 \mu\text{M}$.

Question 3-3.

In the competition FP experiments (Fig 2f) the mP of the F5UR/CRAF-RBD (not BRAF RBD??) is around 220 and with disruption decreases to ~100 mP. Yet in Fig 2d the unbound peptide has a mP of 220. In the competition assay in S2e, the complex starts with a value of 150 mP – why the discrepancy?

The equation used to fit the PF data include the free anisotropy and bound anisotropy values – presumably this should be free and bound polarization values?

Response 3-3:

We thank the reviewer for pointing this out.

We understand that the free probe should essentially have the same mP-value, but these experiments were carried out at very different times and not calibrated as such. Absolute mP-values depend on a number of parameters (buffer composition, temperature, protein size and shape, etc.) and may slightly vary between different measurements. These differences, however, do not affect the maximum of the first derivative and therefore obtained inflection points (K_d, IC₅₀). In these particular cases, it is important to note that differences in the protein batches, the GST-tag (GST-B-RBD vs. C-RBD in **Fig. 2d** and **Fig. S2f**, respectively), and partially automated gain settings can have an impact.

We apologize for the confusion regarding ‘anisotropy’, instead of polarization. The terminology and formula have now been corrected in the Methods (L. 499 following).

To better compare the various binding data, we have summarized them in the **new Table 1**.

Question 3-4.

The mutL5UR removes 4 hydrophobic amino acids, and 2 basic amino acids and replace with 5 acid residues. The L5UR peptide forms a short helix, what impact does incorporation of these mutations have on the structure of the peptide?

Response 3-4:

Structural predictions by AlphaFold2 suggested an unstructured full length L5UR peptide (not shown). Experimental circular dichroism data that are now provided as **new Figure S2g**, suggest that the peptides, L5UR and notably L5URcore and mutL5URcore are similar in structure. They are mostly random coil with ~25% antiparallel β -sheet as the major secondary structure (L. 198). Even though, many peptides only become structured, once bound to their target surface, we no longer argue based on any secondary structure.

Question 3-5.

The effect of the L5UR peptide on the HRAS BRET signal is modest and while the mutant peptide shows no activity is difficult to evaluate the potency. How much would the BRET signal decrease if untagged HRAS-G12V were expressed to compete for the BRET signal.

Jung et al (PMID: 30787043) determined that TRPMK1 attenuates HRAS nanoclustering by disrupting distribution and levels of cholesterol. Inhibitors of TRMK1, such as ML-SI1 show a decrease in HRAS nanoclustering. As Dr. Zhou is an author on both manuscripts it would be interesting to compare the activity of ML-SI1 in the HRAS BRET assay.

Response 3-5:

We thank the reviewer for these excellent suggestions. Both experiments have been performed.

Co-expression of SNAP-tagged H-RasG12V can indeed reduce the BRET-signal by ~85 % as we now demonstrate using an improved BRET-pair with nanoLuc/ mNeonGreen (**new Fig. S3e**). We agree with the more modest effect range of L5UR, which is now compared to the SNAP-H-RasG12V control (L.218):

L5UR or L5UR-SNAP reduced the nanoclustering-BRET by ~33% (Figure 3c), while co-expression of SNAP-H-RasG12V led to a ~85 % reduction (Figure S3e).

By contrast, we did not find any effect of the suggested compound ML-S11 on H-RasG12V-nanoclustering BRET in HEK cells (**Reviewer 3 – Figure 1**). Differences to results found by Jung et al. may stem from the fact that they tested nanoclustering using analysis of electron micrographs of MDCK cell membranes.

In this context, we have also revised the statistical analysis of the electron microscopy nanoclustering data in **Fig. 3d**, which was done using bootstrap testing as now clarified in the Statistics and reproducibility section.

Question 3-6.

The authors show (Fig 1d) and state that Gal1 binds BRAF RBD, yet that use CRAF RBD to demonstrate the activity of L5UR at decreasing the interaction with Gal1 (Fig 2a), but than revert back to GST-BRAF RBD in their fluorescence polarization and pull-down assays. \Figure 2f uses CRAF-RBD (although the methods state BRAF RBD was used) in the FP competition assay. In BRET measurements (Figure 3b) BRAF-RBD is used again, but the modeling of the binding to RBD is done with CRAF. Given the larger signal observed in the Gal1/RBD BRET assay with BRAF RBD and the apparent specificity for BRAF RBD why do the authors switch the source of the RBD in the manuscript? Frankly this just causes confusion and a lack of clarity in the hypothesis being evaluated.

Response 3-6:

We apologize for this confusion, as we intend to demonstrate the breadth of L5UR interactions and the repertoire of data has grown over time starting with C-RBD and later including B-RBD. The breadth of interactions is now even more clear with our new pull-down data showing that L5UR engages a broad range of signaling proteins (**new Fig. 3e**).

We therefore believe it is important to maintain all of these binding data and now provide in the **new Table 1** an overview of mostly low micromolar affinities between L5UR-species and RBDs from Raf-proteins. We have now modified our statement regarding the selective, not ‘specific’ interaction of the L5UR with B-Raf to reflect these data more appropriately.

The structural model of L5URcore/ C-RBD from **Fig. S2d** has been removed, as it was considered too speculative.

Question 3-7.

In BRET experiments measuring HRAS nanoclustering (Fig 2b) the authors show a modest increase in BRET signal with Gal1, the authors should include experiments showing the effect of N-Gal1, to provide further support for their model.

Response 3-7:

We apologize if this has not become clear. In addition to BRET-data in **Fig. 1b**, we also previously provided data in our FRET experiments (**Fig. 2b**), showing that expression of N-Gal1 does not increase the nanoclustering-associated FRET, as it is seen with Gal1 expression, and reported before by us (Blazevits O et al. PMID 27087647).

REVIEWERS' COMMENTS:

Reviewer #1 (Remarks to the Author):

The authors added new experiments to this paper that clarifies the role of the L5UR peptide. It turns out to be not only a disruptor of the Gal1-RBD interface affecting the MAPK pathway, but also appears to have an effect on other pathways such as PI3K and more. Furthermore, the authors pulled back on the structural interpretation of the L5UR interaction and showed by CD experiments that the peptide is unstructured. The authors acknowledge that the L5UR is an immature tool and points out its potential as a starting point for further research. Within this new context, this reviewer's concerns have been addressed.

Reviewer #2 (Remarks to the Author):

The revised manuscript addresses the comments and can be accepted for publication.

Reviewer #3 (Remarks to the Author):

The reviewer would like to acknowledge the changes the authors have made to the manuscript to address the previous concerns. There are a couple of minor corrections/clarifications needed before the manuscript can be published:

1. Line 190-1, "L5URcore could displace F-L5UR from the C-RBD with a slightly reduced potency ($IC_{50} = 14 \pm 6 \mu M$) (Figure 2e,f)." Should this IC_{50} refer to Figure S2f and not Figure 2e, f?
2. In Figure 4b the x-axis seems elongated between -5 and -4 \log_{10} . Could the authors check this?

Point-by-point response to revision on manuscript COMMSBIO-23-3781A: "Identification of an H-Ras nanocluster 1 disrupting peptide"

Reviewer #1:

The authors added new experiments to this paper that clarifies the role of the L5UR peptide. It turns out to be not only a disruptor of the Gal1-RBD interface affecting the MAPK pathway, but also appears to have an effect on other pathways such as PI3K and more. Furthermore, the authors pulled back on the structural interpretation of the L5UR interaction and showed by CD experiments that the peptide is unstructured. The authors acknowledge that the L5UR is an immature tool and points out its potential as a starting point for further research. Within this new context, this reviewer's concerns have been addressed.

Response to #1:

We are grateful to the reviewer for helping us to improve our manuscript.

Reviewer #2:

The revised manuscript addresses the comments and can be accepted for publication.

Response to #2:

We thank the reviewer for their feedback and support for acceptance.

Reviewer #3:

The reviewer would like to acknowledge the changes the authors have made to the manuscript to address the previous concerns. There are a couple of minor corrections/clarifications needed before the manuscript can be published:

1. Line 190-1, "L5URcore could displace F-L5UR from the C-RBD with a slightly reduced potency ($IC_{50} = 14 \pm 6 \mu M$) (Figure 2e,f)." Should this IC_{50} refer to Figure S2f and not Figure 2e, f?
2. In Figure 4b the x-axis seems elongated between -5 and -4 \log_{10} . Could the authors check this?

Response to #3:

Thank you for your thorough review and assessment. The minor corrections have been made, our apologies for these mistakes.

Specifically:

1. Indeed, the reference needed to be corrected to Figure S2f: '[...] L5URcore could displace F-L5UR from the C-RBD with a slightly reduced potency ($IC_{50} = 14 \pm 6 \mu M$) (Table1, Supplementary Fig. 2f).'
2. This mistake in Figure 4b, which apparently occurred when not dragging all drawing elements within our graphing software, has been corrected.